# Genome-Wide Identification of the RsSWEET Gene Family and Functional Analysis of *RsSWEET17* in Root Growth and Development in Radish

Xiaoli Zhang [1,†], Yang Cao [1,†], Ruixian Xin [1], Liang Xu [1], Yan Wang [1], Lun Wang [2], Yinbo Ma [2] and Liwang Liu [1,2,*]

1  National Key Laboratory of Crop Genetics & Germplasm Enhancement and Utilization, Key Laboratory of Horticultural Crop Biology and Genetic Improvement (East China) of MOAR, College of Horticulture, Nanjing Agricultural University, Nanjing 210095, China; 2020204030@njau.edu.cn (X.Z.); 2016104069@njau.edu.cn (Y.C.); 2021104062@stu.njau.edu.cn (R.X.); nauxuliang@njau.edu.cn (L.X.); wangyanhs@njau.edu.cn (Y.W.)
2  College of Horticulture and Landscape Architecture, Yangzhou University, Yangzhou 225009, China; wanglun@yzu.edu.cn (L.W.); mayb@yzu.edu.cn (Y.M.)
*  Correspondence: nauliulw@njau.edu.cn
†  These authors contributed equally to this work.

**Abstract:** SWEET (*Sugars Will Eventually be Exported Transporter*) genes play essential roles in various biological processes, including phloem loading, sugar efflux, plant development and stress response. In this study, a total of 33 *RsSWEET* gene members were identified in the radish genome. They could be divided into four subfamilies and are distributed on eight radish chromosomes. *Cis*-acting regulatory element analysis indicated that these *RsSWEET* genes were potentially involved in the radish growth and development and stress response process, including circadian control and light response and responses to numerous stresses, including low-temperature and drought stress. Transcriptome data analysis revealed that a number of *RsSWEET* genes exhibited specific expression patterns in different tissues and developmental stages of radish. Moreover, several *RsSWEET* genes (e.g., *RsSWEET2a*, *RsSWEET3a*, *RsSWEET16b* and *RsSWEET17*) showed differential expression profiles under various abiotic stresses, including cold, heat, salt, Cd and Pb stress. Remarkably, the *RsSWEET17* was specifically expressed in the cambium of radish. *RsSWEET17* was heterologously expressed in yeast strain EBY.VW4000, which suggested that it has the ability to transport sugar. Notably, *RsSWEET17*-overexpressing *Arabidopsis* plants exhibited excessive root length, greater fresh weight and higher soluble sugar content (SSC) accumulation compared with wild-type (WT) plants, indicating that *RsSWEET17* might positively regulate radish taproot development by strategically manipulating sugar accumulation. Collectively, these results clarify the molecular mechanisms underlying *RsSWEET*-mediated sugar accumulation and root growth and development in radish.

**Keywords:** radish; SWEET; stress response; *RsSWEET17*; sugar transport activity; sugar accumulation

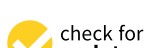



## 1. Introduction

Sugars are a crucial primary energy and carbon source for plant growth and development [1,2]. In plants, sugars are synthesized during photosynthesis in leaves or other source organs and then transported via the phloem to various sink tissues and organs such as roots, shoots and seeds [3,4]. Sugars are loaded via the symplastic pathway for transport in the phloem [5], and sugar transport requires sugar transporter proteins [6]. Three eukaryotic sugar transporter families have been extensively proven to act as sugar/H$^+$ symporters and play critical roles in sugar translocation and distribution, which significantly determine crop yield and quality [7–9].

Sucrose transporters (SUTs), belonging to the major facilitator superfamily, play pivotal roles in mediating the long-distance sucrose transport from source to sink [10,11]. Monosaccharide transport is mainly mediated by MST transporter proteins [12]. SWEETs are a family of sugar efflux transporters that greatly facilitate the diffusion of sugars across intracellular or plasma membranes [13,14]. *SWEET* genes have been retrieved from a variety of plants, including *Arabidopsis* [15], *Brassica rapa* [16], tomato (*Solanum lycopersicum*) [17] and rice [18]. Structural analysis revealed that the SWEET proteins in eukaryotes harbor two MtN3/saliva domains, and the SWEET gene family in plants could be classified into four clades [15,19]. Further studies revealed that *SWEET* genes greatly affect the transportation, metabolism and storage of carbohydrates in plants [20,21]. *AtSWEET11* and *AtSWEET12* are the key players of sucrose efflux [6], and *AtSWEET17* regulates the sugar content in roots and leaves [22,23]. Overexpression of *SlSWEET1a* and *PuSWEET15* leads to increased soluble sugar content in tomato and pear, respectively [19,24]. Increasing evidence has demonstrated that *SWEET* genes are also involved in the response to environmental stress, including heat, cold and salt stress [15,25–27]. *OsSWEET13* and *OsSWEET15* are involved in maintaining sugar homeostasis under high-salinity stress in rice [28]. Overexpression of *MdSWEET16* enhanced cold tolerance by increasing sugar accumulation in apples [29]. These results indicate that *SWEET* genes play a crucial role in plant growth as well as various abiotic responses in plants.

Radish (*Raphanus sativus*) is an economically important root vegetable with high nutritional value [30]. The taproot makes up an essential part of the edible radish. The accumulation of carbohydrates is beneficial for taproot development. Carbon assimilates are products of photosynthesis and are transported to sink organs to support plant development [31]. *SWEETs* play crucial roles in multiple processes, including carbohydrate transportation and plant growth and development [6]. It is feasible to improve taproot development through genetic manipulation of the *SWEET* gene involved in sucrose allocation. Therefore, the comprehensive identification and functional analysis of *SWEET* genes is of great importance for exploring the genetic and molecular basis of excellent traits in plants. Although the molecular characterization of *SWEETs* has been investigated in several plant species, the systematic identification and molecular function analysis of SWEET genes remains largely unexplored in radish. In this study, we mainly focused on the identification of all *RsSWEET* genes from the radish genome, while exploring the phylogenetic relationships, gene structures, conserved domains and *cis*-acting elements of the *RsSWEET* gene family in radish. In addition, the expression profiles of *RsSWEET* genes were investigated in different tissues and developmental stages, and the differential expression of eight *RsSWEET* genes (*RsSWEET2a*, *RsSWEET3a*, *RsSWEET11a/b*, *RsSWEET12a/b*, *RsSWEET16b* and *RsSWEET17*) was assessed under various abiotic stresses with RT-qPCR analysis. Furthermore, the potential roles of *RsSWEET17* in sugar accumulation and taproot development were investigated in yeast and radish. The findings provide insights into the molecular basis underlying the involvement of *RsSWEET* genes in taproot development and stress response in radish plants and facilitate genetic improvement of vital horticulture traits in breeding programs.

## 2. Materials and Methods

### 2.1. Identification of SWEET Genes in Radish

The sequences of radish SWEET proteins were downloaded from the 'NAU-LB' radish genome (Bioproject number: PRJCA011486) [30]. To confirm the predicted genes, the seed sequence (PF03083, http://pfam.xfam.org/; accessed on 15 December 2022) with the MtN3/saliva domain was searched against the radish genome using HMMER 3.0 and the BLASTP program with an E-value set to 0.01 [32,33]. Each sequence was submitted to Pfam (http://pfam-legacy.xfam.org/; accessed on 20 December 2022) and SMART (https://smart.embl.de/; accessed on 25 December 2022) for further verification.

## 2.2. Phylogenetic Analysis and Gene Structure Analyses of the RsSWEET Genes

The SWEET protein sequence of *Arabidopsis* was downloaded from TAIR10 (https://www.arabidopsis.org/; accessed on 5 January 2023). The SWEET protein sequence of *Brassica rapa* was obtained from the BRAD database (http://brassicadb.org/brad/; accessed on 5 January 2023). MUSCLE was used to perform multiple alignments of the full-length SWEET protein sequences of *A. thaliana*, *B. rapa* and *R. sativus* with default parameters. Subsequently, the phylogenetic tree was constructed using MEGA 6.0 via the neighbor-joining (NJ) method with 1000 bootstrap replicates and the Poisson model [34,35]. The collinearity analysis of *SWEET* genes between *A. thaliana*, *B. rapa* and *R. sativus* was performed using the MCScanX and TBtools v1.120 [36]. In addition, *RsSWEET* genes were mapped to eight chromosomes using TBtools v1.120 software based on the corresponding location parameters in the NAU-LB radish genome [30]. The Gene Structure Display Server (GSDS) (http://gsds.gao-lab.org/; accessed on 10 February 2023) was used to analyze the gene structure [37]. The MEME tool (https://meme-suite.org/meme/doc/meme.html; accessed on 20 February 2023) was employed to identify conserved motifs [38].

## 2.3. Promoter Cis-Element Analysis of RsSWEETs

The PlantCARE (https://bioinformatics.psb.ugent.be/webtools/plantcare/html/; accessed on 8 March 2023) database was used to identify potential *cis*-elements in the promoter sequences of *RsSWEET* genes (1.5 kb upstream of the translation start site) [39]; the promoter sequences were obtained from the NAU-LB radish genome.

## 2.4. Plant Materials and Treatments

The radish inbred line 'NAU-XBC' was employed in this study. The germinated seeds were sown in soil with 14 h light (25 °C) and 10 h dark (16 °C). Radish plants in the taproot-thickening period were selected for stress treatment. For cold treatment, radish plants were treated at 4 °C for 0 h, 6 h, 12 h, 24 h and 7 days. For heat stress, radish plants were treated at 40 °C for 0 h, 4 h, 8 h, 12 h and 24 h. For salt stress, these plants were treated with 250 mM NaCl for 0 h, 6, 12, 24 and 48 h. Radish was irrigated with $CdCl_2$ (Cd, 50 μM) and $Pb(NO_3)_2$ (Pb, 100 μM) stress at 0 h, 12 h, 24 h, 48 h and 7 days. Root samples were collected after various stress treatments and stored at −80 °C for use.

The *35S::RsSWEET17* fusion plasmid constructs were transferred into *Arabidopsis* plants with the floral dip method [30]. The positive transgenic $T_1$ seedlings were screened to generate the $T_3$ transgenic lines which were further utilized for the phenotypic analyses.

## 2.5. Expression Profile Analysis of RsSWEET Genes

Previous RNA-seq data were retrieved to analyze the expression level of the *RsSWEET* genes in four different tissues (leaf, root, pistil and stamen) (BioProject number: PRJCA011507, National Genomics Date Center) [30] and in three tissues (Ca: cambium, Co: cortex, Pa: parenchyma) and stages (5/7/9 weeks) of radish taproot (GenBank: PRJNA475856) [40]. The expression level of *RsSWEETs* was displayed by the RPKM value, and the heat map was generated using TBtools v1.120 software.

## 2.6. Functional Complementation Analyses in Yeast

The CDS sequence of *RsSWEET17* was fused on the pDR196 vector to produce the pDR196-*RsSWEET17* fusion expression vectors. Then, the plasmids of pDR196 and pDR196-*RsSWEET17* were transferred into EBY.VW4000 yeast [41], employing the lithium acetate method [2]. The pDR196 vector was used as a control. EBY.VW4000 belongs to the group of hexose-deficient yeast, which can only grow on a medium with maltose as the sole carbon source. Transformants were grown on SD/- Ura solid medium with 2% maltose for 3 days. For function assays, single yeast cells were grown in SD/- Ura liquid medium overnight. Then, the yeast was adjusted to an optical density at 600 nm (OD600) of 1.0 and then diluted with 10×, 100× and 1000×. In total, 4 μL diluent was plated on SD/- Ura solid media with

2% maltose (as control), 2% glucose, 2% fructose, 2% mannose or 2% galactose. The culture plates were incubated at 30 °C for 3–5 days. The primers are listed in Table S4.

### 2.7. Soluble Sugar Determination

Soluble sugars were extracted and determined as described previously [30]. In brief, after growing WT and transgenic *Arabidopsis* plants on MS solid medium for 20 days, 0.2 g fresh weight was ground after being frozen in liquid nitrogen. The samples were dissolved in 1.5 mL extraction solution (80% ethanol) for 30 min (80 °C). The content of soluble sugars was analyzed with UPLC [33,42]. The total sugar contents contain sucrose, glucose and fructose content.

### 2.8. RNA Isolation and RT-qPCR

The total RNA of each sample was extracted and cDNA was synthesized according to the reported method [30]. The expression levels of *RsSWEETs* were examined in response to various stresses by qRT-PCR on a LightCycler 480 System (Roche, Mannheim, Germany). The relative expression ratio was calculated with the $2^{-\Delta\Delta C_T}$ method, and the *RsActin* gene was selected as an internal control [43]. All primers for RT-qPCR analysis are shown in Table S3.

### 2.9. Statistical Analysis

Data were analyzed using SPSS statistical software (SPSS 20.0, IBM, Chicago, IL, USA). Student's *t*-test was used to determine statistical significance. All values are represented as mean ± standard error (SE).

## 3. Results

### 3.1. Identification, Phylogenetic and Collinearity Analysis of RsSWEETs in Radish

In this study, 33 SWEET proteins were identified from the radish genome and were named RsSWEET1 to RsSWEET17 according to the similarity to these SWEET proteins in Arabidopsis (Table S1). According to the classification of *Arabidopsis*, the RsSWEET gene family can be divided into four subfamilies, AI, AII, AIII and BI. In addition, an unrooted phylogenetic tree was constructed by aligning 33, 17 and 30 SWEET proteins from *R. sativus*, *A. thaliana* and *B. rapa*, respectively, further verifying the phylogenetic relationships and classification of RsSWEET proteins (Figure 1, Table S1). Briefly, 13 of the 33 *RsSWEET* genes (39.39%) belonged to the BI subfamily, which was the most common member of the radish SWEET family. There were 10, 3 and 7 *RsSWEETs* belonging to the subfamilies AI, AII and AIII, accounting for 33.33%, 9.09% and 21.21% of the total *RsSWEETs*, respectively. Subsequently, multiple alignments for SWEET protein sequences of *A. thaliana*, *B. rapa* and *R. sativus* revealed that RsSWEET contained the characteristic MtN3/saliva domain (Figure S1).

To further explore the collinearity analysis of *SWEET* genes, the syntenic map was constructed based on the collinearity relationships among *R. sativus*, *A. thaliana* and *B. rapa* genomes (Figure 2, Table S2). The results showed that there were 69 syntenic orthologous gene pairs between 33 *RsSWEETs* and 30 *BrSWEETs*. Of these, 13 *RsSWEETs* have 3 corresponding orthologous genes in *B. rapa*, such as *RsSWEET3a/RsSWEET3b/RsSWEET3c-BrSWEET3*, indicating these genes had a common ancestor. Additionally, 30 RsSWEETs have a syntenic relationship with 15 *AtSWEET* orthologous gene pairs between *R. sativus* and *A. thaliana*. Most *RsSWEET* genes had a single corresponding orthologous *AtSWEET* gene. The results indicated that a proportion of *RsSWEET* genes have more homologous genes in *B. rapa* than in *A. thaliana*.

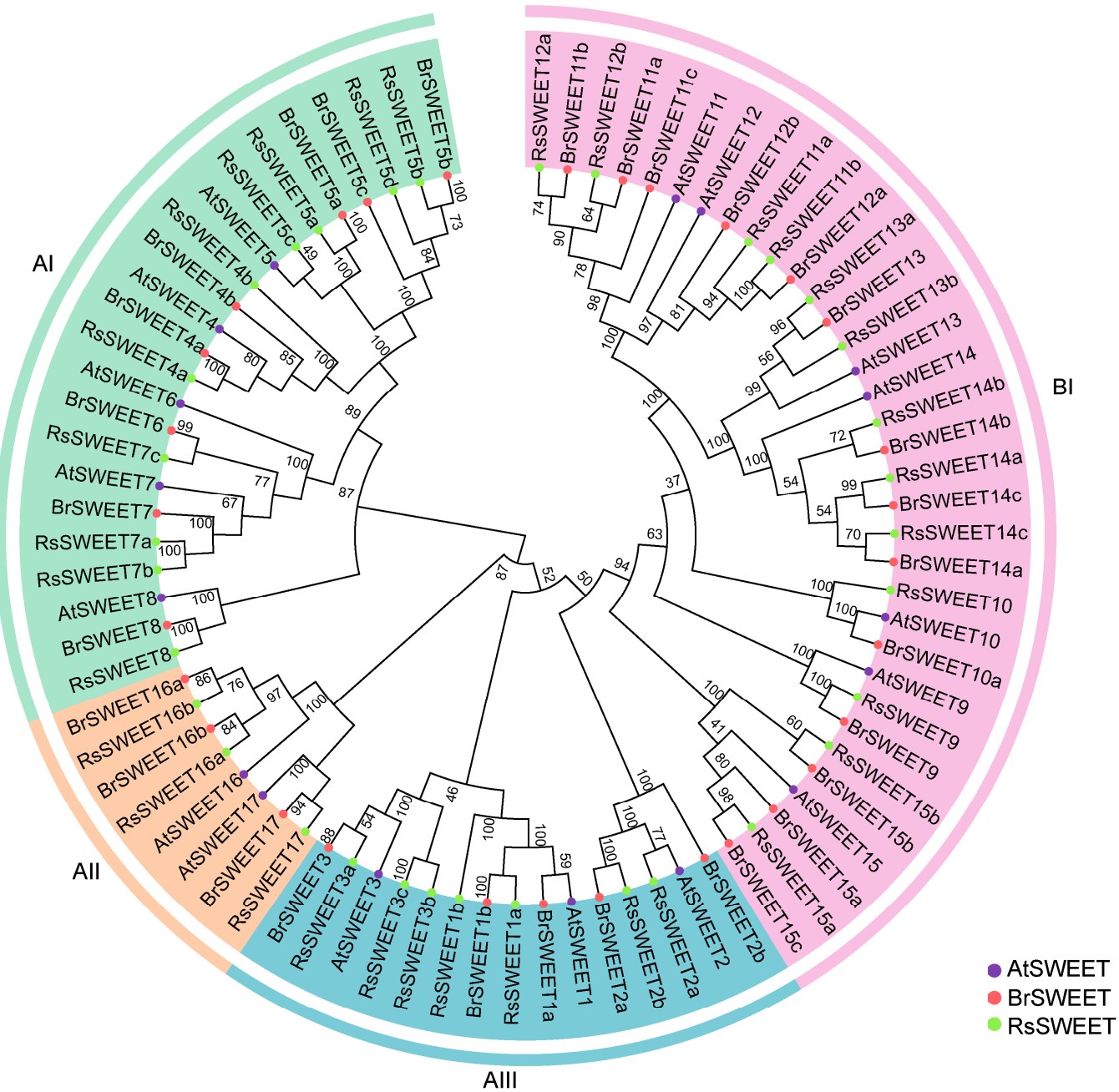

**Figure 1.** Phylogenetic relationships of SWEET proteins from *A. thaliana*, *B. rapa* and *R. sativus*. Subfamilies are marked in different colors. At: *A. thaliana*, Br: *B. rapa*, Rs: *R. sativus*.

### 3.2. Chromosomal Distribution of RsSWEET Proteins in Radish

In total, 33 *RsSWEET* genes were distributed on 8 chromosomes of radish. In general, the distribution of these *RsSWEET* genes on chromosomes was relatively scattered (Figure 3). Among them, more than half of the genes were distributed on the chromosomes Chr4, Chr6 and Chr9, which contained six genes. There were four members located on each of the chromosomes Chr2, Chr3 and Chr5, whereas chromosome Chr8 harbored two members. Only one gene was found on chromosome Chr1. No *RsSWEET* genes were detected on Chr7. In addition, two pairs of genes (*RsSWEET11a/b* and *RsSWEET5b/d*) on the chromosome were tightly linked.

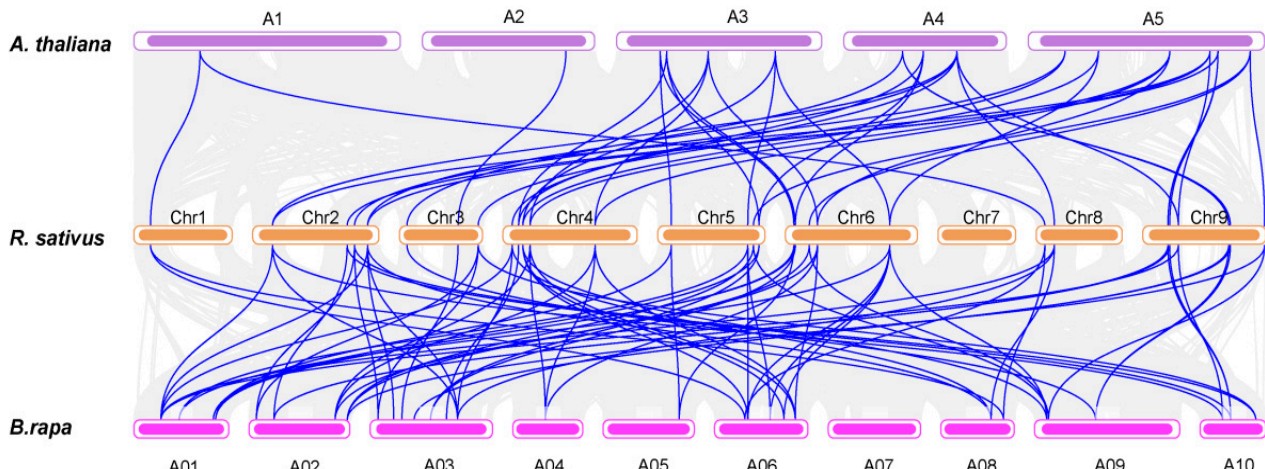

**Figure 2.** The collinearity analysis of *SWEET* genes among *R. sativus*, *A. thaliana* and *B. rapa*. The gray line indicates the collinearity among all genes in the genomes, and the blue line indicates the *SWEET* gene pairs.

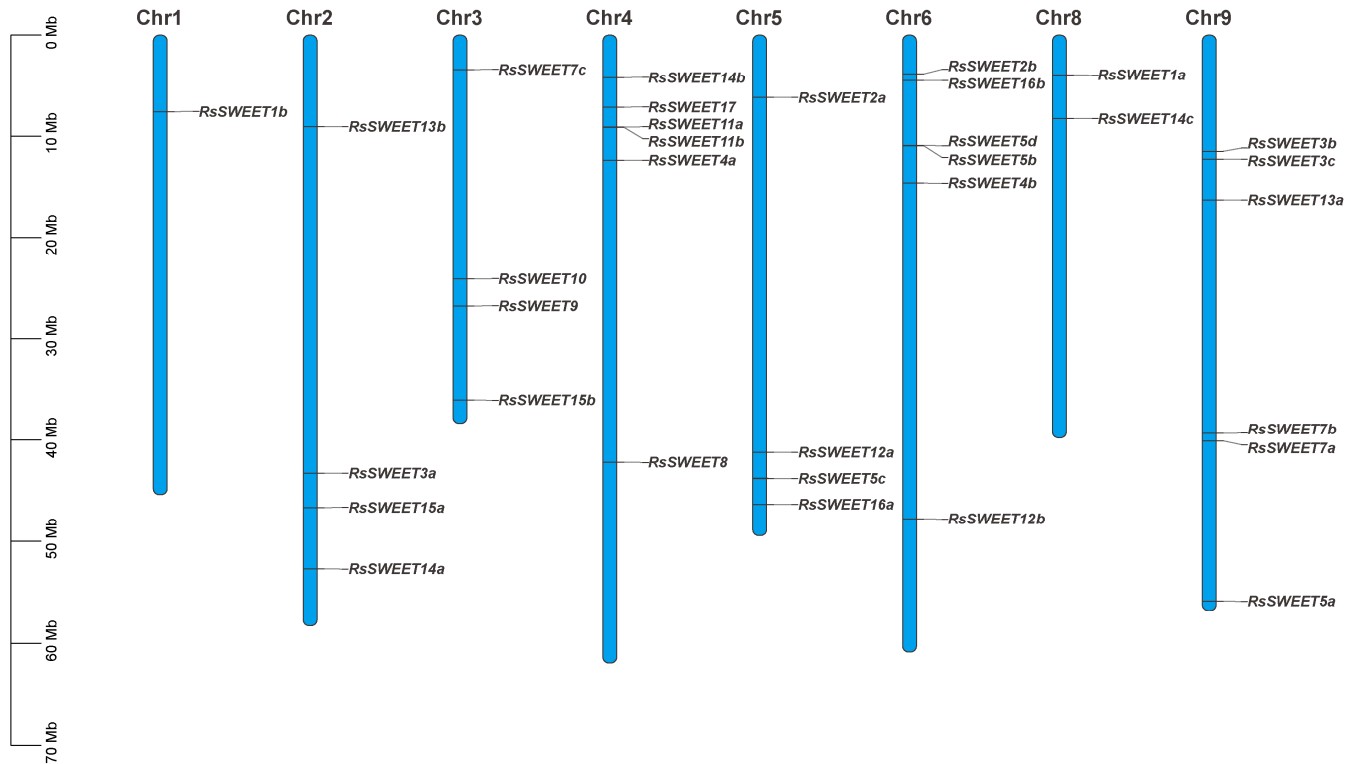

**Figure 3.** The chromosomal localization of the RsSWEET family members in the radish genome.

### 3.3. Gene Structure and Conserved Motif Distribution of RsSWEETs

To further investigate the structural features of *RsSWEET* genes, the intron–exon structure was displayed (Figure 4A,B). The gene structure of the same branch members of the SWEET gene family was similar, except for the difference in exon and intron length. Overall, 27 members (51.6%) contained 6 exons, 3 members (19.35%) contained 5 exons and 2 members (9.68%) contained 4 exons. Moreover, there were seven exons in *RsSWEET16a*. Eleven putative conserved motifs in all 33 RsSWEET proteins were predicted using the MEME program (Figure 4C). The results revealed that members in the same subfamily contained similar motifs, suggesting that each subfamily protein probably exerts similar functions. Motifs 1, 3 and 5 were distributed in all proteins, indicating that these motifs were highly conserved. Nevertheless, several motifs were only presented in some specific

subfamilies; for instance, motif 9 was observed in AI and BI subfamilies, and motif 10 was only present in AIII, whereas motifs 7 and 8 were only detected in BI subfamilies.

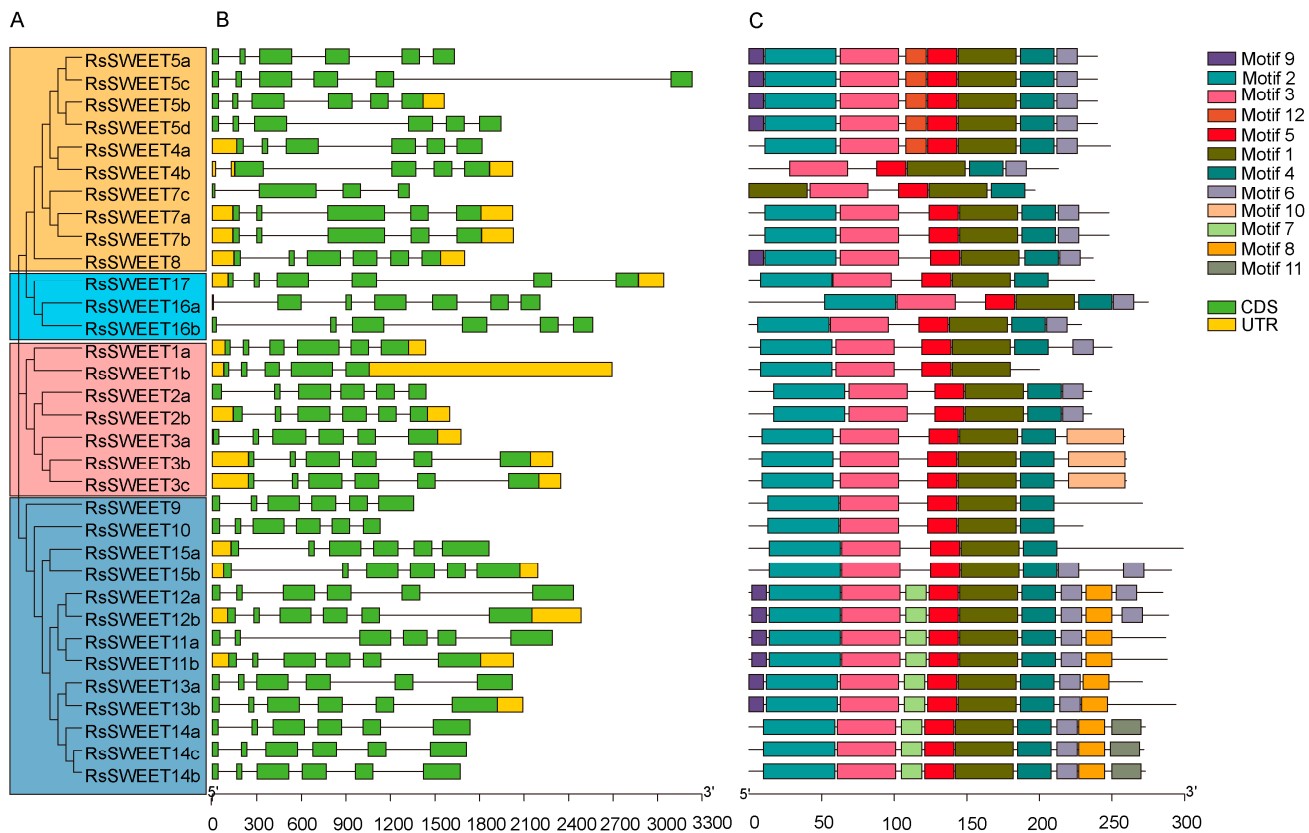

**Figure 4.** The analysis of RsSWEET protein and gene structure. (**A**) Phylogenetic relationship of RsSWEET protein. Different colors represent different subfamilies of RsSWEET gene family. (**B**) Gene structure of *RsSWEET* genes. (**C**) Conserved motifs of RsSWEET proteins.

### 3.4. The Cis-Acting Regulatory Elements in RsSWEET Gene Promoters

The transcriptional regulation of *RsSWEETs* was analyzed using the PlantCARE server. In this study, a variety of CREs in the *RsSWEET* gene promoters were associated with light response, including ACE, G-box, GA-motif, GT1-motif and MRE, indicating that the *RsSWEETs* may be regulated by light and participate in photosynthesis. In addition, the top ten elements related to various stresses and hormone responses were identified in the promoter of *RsSWEETs* (Figure 5 and Table S3), such as the low-temperature response element (LTR), the heat shock element (HSE), ABA response (ABRE), salicylic acid (TCA-element) and auxin (TGA-element). Moreover, some elements related to growth and development, including meristem expression (CAT-box) and endosperm expression (GCN4-motif), were identified in *RsSWEET5a*, *RsSWEET16b* and *RsSWEET17*. The *cis*-acting element distribution and numbers of *RsSWEETs* varied significantly. For instance, the promoter sequence of *RsSWEET11b* and *RsSWEET12a* has a higher number of stress-responsive *cis*-elements, while *RsSWEET5d* and *RsSWEET16b* carried a higher number of hormone response-related cis-elements. It could be suggested that *SWEET* genes contain various elements which play a vital role in plant development and stress response.

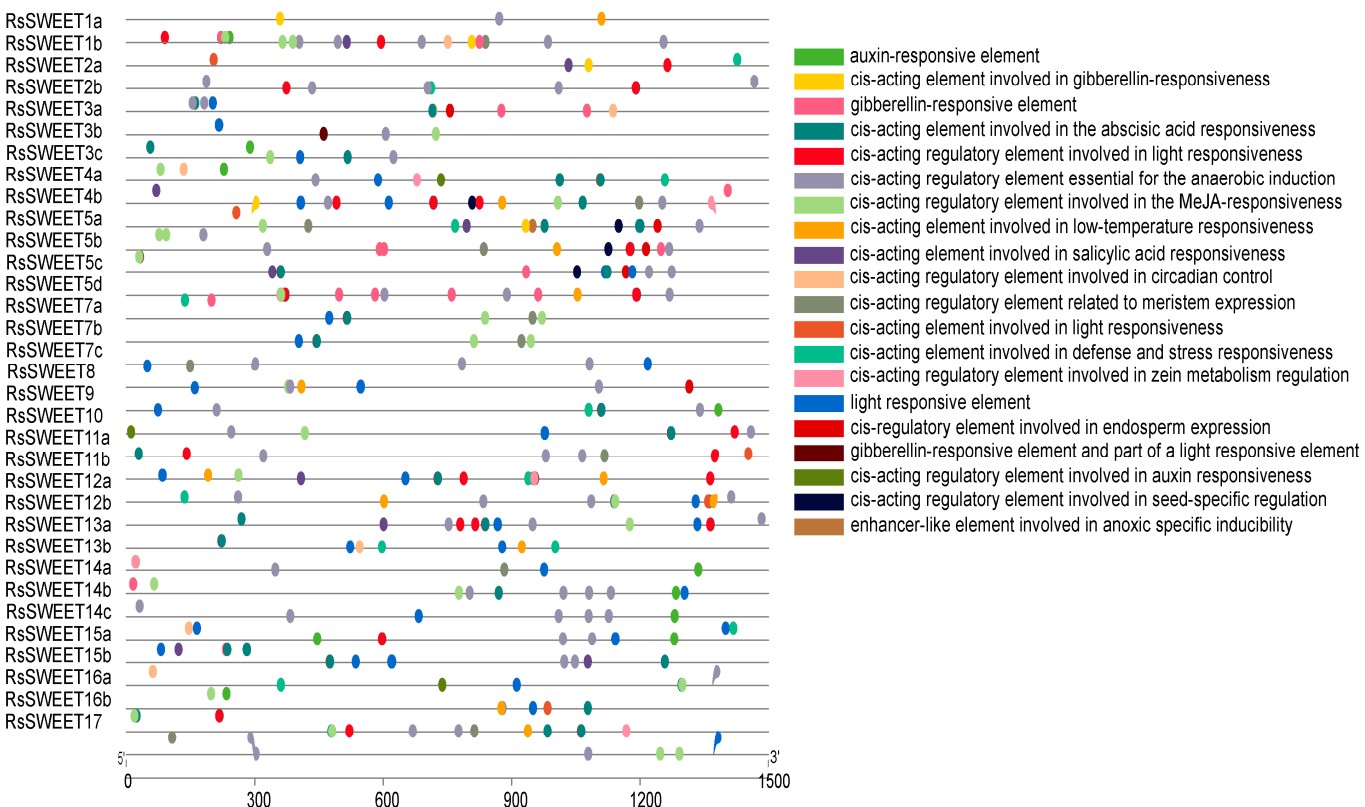

**Figure 5.** *Cis*-regulatory elements in the promoter regions of the *RsSWEET* genes. The different colored boxes represent different *cis*-regulatory elements.

### 3.5. Differential Expression Analysis of RsSWEET Genes

To investigate the expression levels of *RsSWEET* genes, their gene expression profiles in four different tissues and three development stages were determined (Figure 6). The results showed that *RsSWEETs* exhibited tissue- or developmental stage-specific expression varying in the course of development in radishes. Overall, *RsSWEET11b*, *RsSWEET12a/b* and *RsSWEET13b* in the BI group were highly expressed in leaves, while *RsSWEET13b*, *RsSWEET14a/c* and *RsSWEET15a/b* were highly expressed in stamens. Similarly, *RsSWEET4b*, *RsSWEET5b/c/d* and *RsSWEET8* in AI showed a high expression level in stamens but were rarely expressed in roots. Moreover, there was a slight difference in expression between AII and AIII members except for *RsSWEET3a* in roots. Interestingly, *RsSWEET16b* and *RsSWEET17* were highly expressed in roots but were hardly expressed in other tissues, suggesting that these genes might be root-specific and play important roles in root growth (Figure 6A).

The expression patterns of *RsSWEET* genes varied at different developmental stages of the radish taproot (Figure 6B). Generally, the expression of BI genes was stable in the cambium, cortex and parenchyma but not high, except *RsSWEET11a/b*, *RsSWEET12a/b*, *RsSWEET14c* and *RsSWEET15a*. In the subfamilies AI and AIII, *RsSWEET1b*, *RsSWEET2a/b* and *RsSWEET4a/b* showed a high expression level in the cambium region at 7 weeks of radish growth. Meanwhile, the *RsSWEET3a/c* exhibited a high value in the parenchyma region at 9 weeks of radish growth. Additionally, *RsSWEET17* in AII is specifically expressed in the cambium region and changes with the development stages of radish, suggesting it might play a crucial role in regulating cambium activities and improvement of taproot yield.

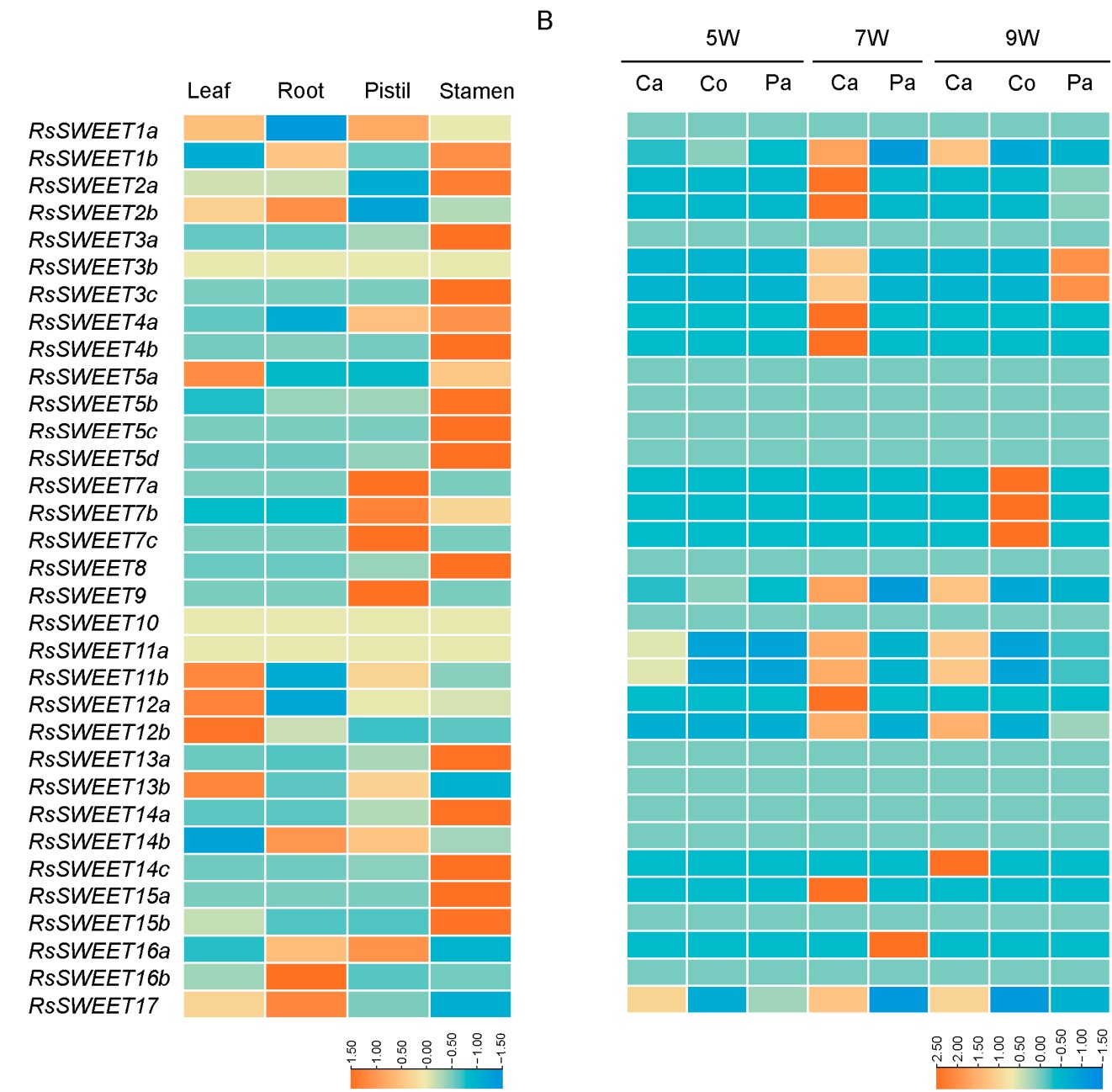

**Figure 6.** Expression profile analysis of *RsSWEETs* in different tissues and stages of radish plants. (**A**) The expression profiles of *RsSWEETs* in four tissues (root, leaf, pistil and stamen). (**B**) A heat map of *RsSWEET* expression in three tissues (Ca, cambium; Co, cortex; Pa, parenchyma) at three growth stages (W, weeks). The log$_2$ (FPKM) value was used for the expression level of *RsSWEETs*.

*3.6. Expression Profiles of RsSWEET Genes under Abiotic Stress in Radish*

Total eight *RsSWEETs* (*RsSWEET2a*, *RsSWEET3a*, *RsSWEET11a/b*, *RsSWEET12a/b*, *RsSWEET16b* and *RsSWEET17*) were highly expressed in the cambium regions of radish taproot, indicating that they might play an important role in source–sink carbohydrate partitioning and organ development. To investigate the dynamic expression patterns of *RsSWEET* genes under various abiotic stresses in radish, these eight *RsSWEETs* were selected for RT-qPCR assays. Some *SWEET* genes, such as *RsSWEET2a*, *RsSWEET3a*, *RsSWEET11b*, *RsSWEET12a* and *RsSWEET17*, were induced by cold stress and reached a maximum level at 12 h (Figure 7A). The gene expression level of *RsSWEET2a* and *RsSWEET3a* reached a maximum level at 7 days, suggesting that they might be greatly involved in response to

long-term cold stress. *RsSWEET11b* exhibited upregulation in expression under short-term cold stress. Meanwhile, these *RsSWEET* genes, excluding *RsSWEET11b*, were also induced under heat stress (Figure 7B). The *RsSWEET3a* and *RsSWEET11b* genes responded preferentially to heat stress compared to *RsSWEET2a* and *RsSWEET11a/b*, while *RsSWEET16b* and *RsSWEET17* were strongly involved in long-term heat stress. Consistently, relative to untreated radish plants, these *RsSWEET* genes also were induced under salt, Cd and Pb stress (Figure S2). Collectively, the *RsSWEET17* gene exhibited a high expression level in radish taproot and was triggered under abiotic stress, indicating a positive pleiotropic effect on the taproot growth and response to stresses. The *RsSWEET17* gene was further investigated as a candidate gene for radish taproot development.

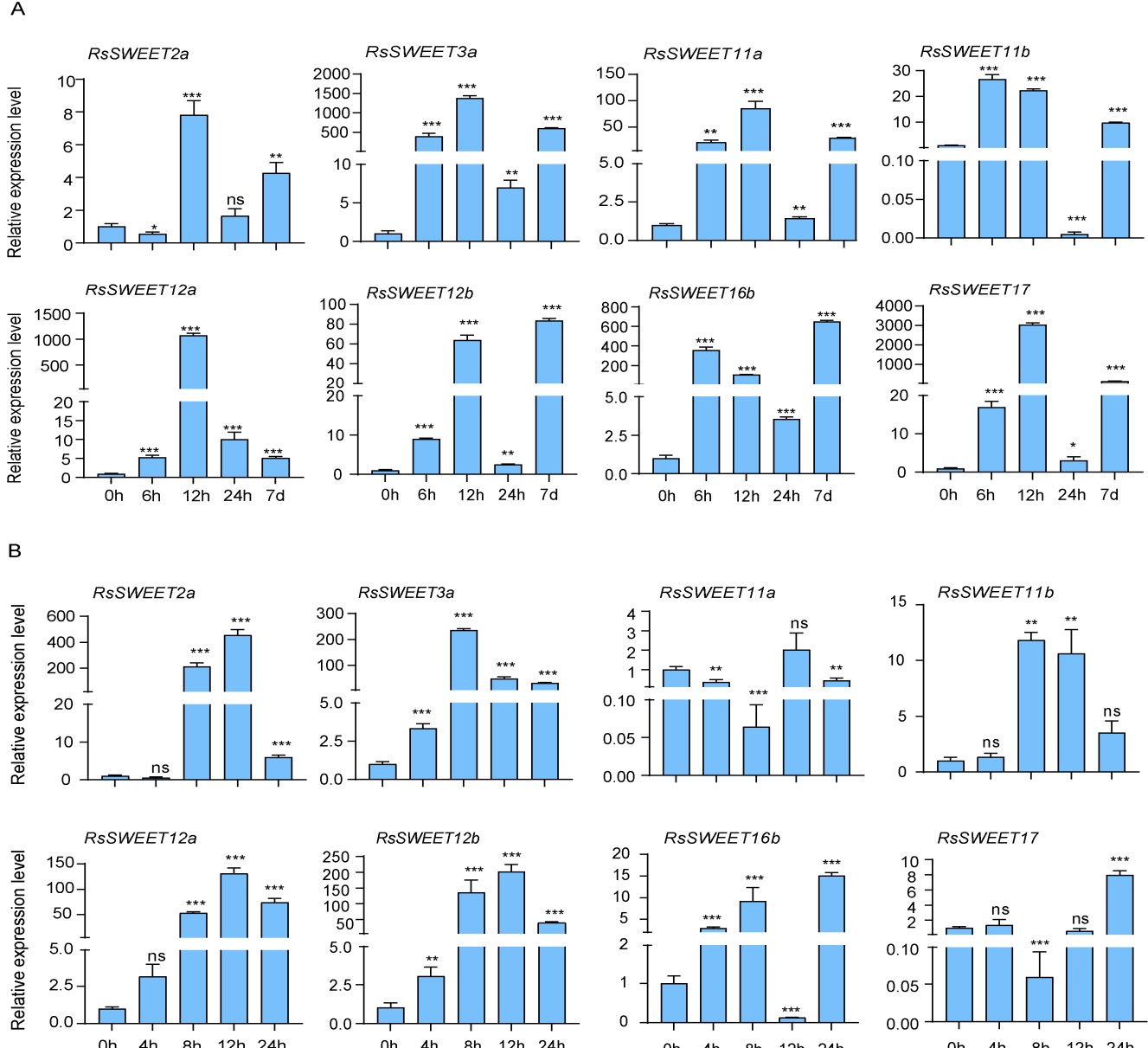

**Figure 7.** RT-qPCR expression analysis of *RsSWEET* genes under cold and heat stresses. Treatment time (h, hour; d, day). (**A**) For cold treatment, radish plants were treated at 4 °C for 0 h, 6 h, 12 h, 24 h and 7 days. (**B**) For heat stress, radish plants were treated at 40 °C for 0 h, 4 h, 8 h, 12 h and 24 h. The asterisks above bars indicate a statistically significant difference with Student's *t*-test (ns, not significant; * $p < 0.05$; ** $p < 0.01$; *** $p < 0.001$).

### 3.7. Sugar Transport Activity of RsSWEET7 in Yeast

To investigate the sugar transport ability of RsSWEET17, the fusion vector pDR196-RsSWEET17 was expressed in the cells of yeast strain EBY.VW4000. As shown in Figure 8, compared with the pDR196 vector transformant, the yeast transformant of RsSWEET17 showed recovery in media containing only glucose, fructose, galactose or mannose. This result indicated that the deficiency of hexose uptake can be supplemented by RsSWEET17 in yeast.

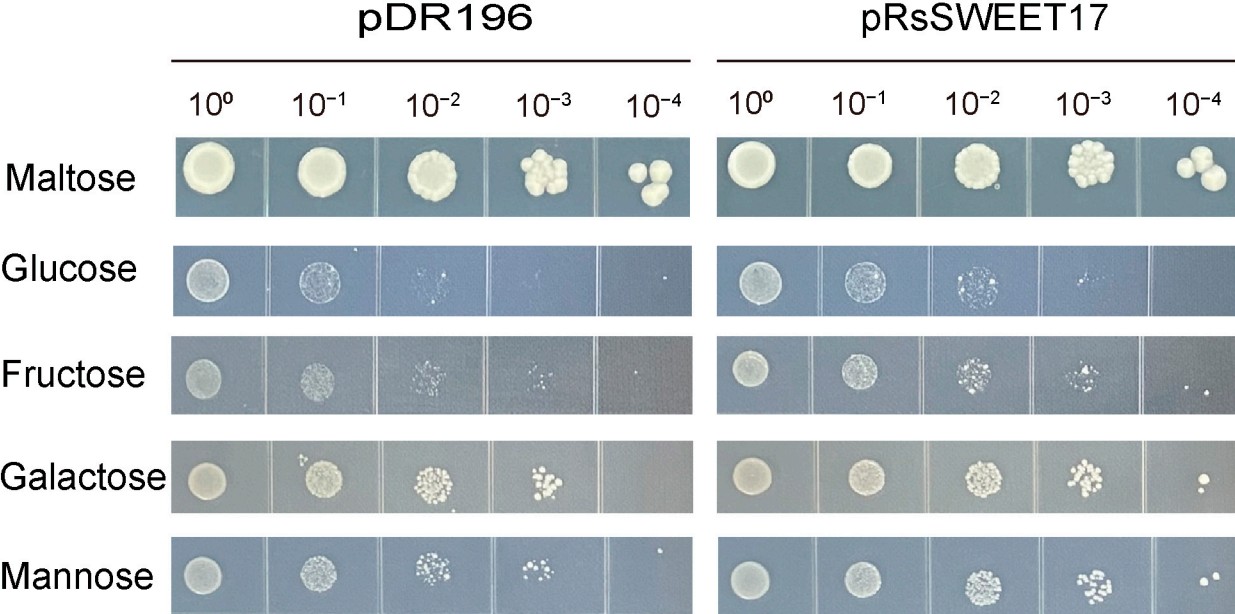

**Figure 8.** Sugar transport activity analysis of RsSWEET17 in EBY.VW4000 yeast cells. Transformants were grown on SD/- Ura solid medium containing 2% maltose, glucose, fructose, galactose or mannose independently for 3 days.

### 3.8. Overexpression of RsSWEET17 Improves Root Sugar Accumulation and Root Growth

To further identify the role of *RsSWEET17* in sugar accumulation, transgenic $T_3$ *Arabidopsis* plants overexpressing *RsSWEET17* (*RsSWEET17-OE*#9 and *RsSWEET17-OE*#12) were generated (Figure 9A). It was obvious that the transgenic lines overexpressing *RsSWEET17* had much-improved root growth compared with the WT plants in the MS solid medium for 20 days (Figure 9A). The *RsSWEET17* gene exhibited higher expression in the roots and leaves of two independent $T_3$ transgenic lines (Figure 9B). The root length of *OE*#9 and *OE*#12 transgenic plants was 1.24- and 1.22-fold greater than that of the WT plants, respectively (Figure 9C). The fresh weight of *OE*#9 and *OE*#12 transgenic plants was greater than that of WT plants (Figure 9D). Furthermore, the total soluble sugar content (TSSC) of the $T_3$ transgenic lines was higher than that of WT plants (Figure 9E). In particular, for two independent $T_3$ transgenic lines, the sucrose (Suc) content and the glucose (Glc) increased by 1.5–1.6 and 2.6–2.7 fold, respectively, while fructose (Fru) changed slightly compared to the wild plants.

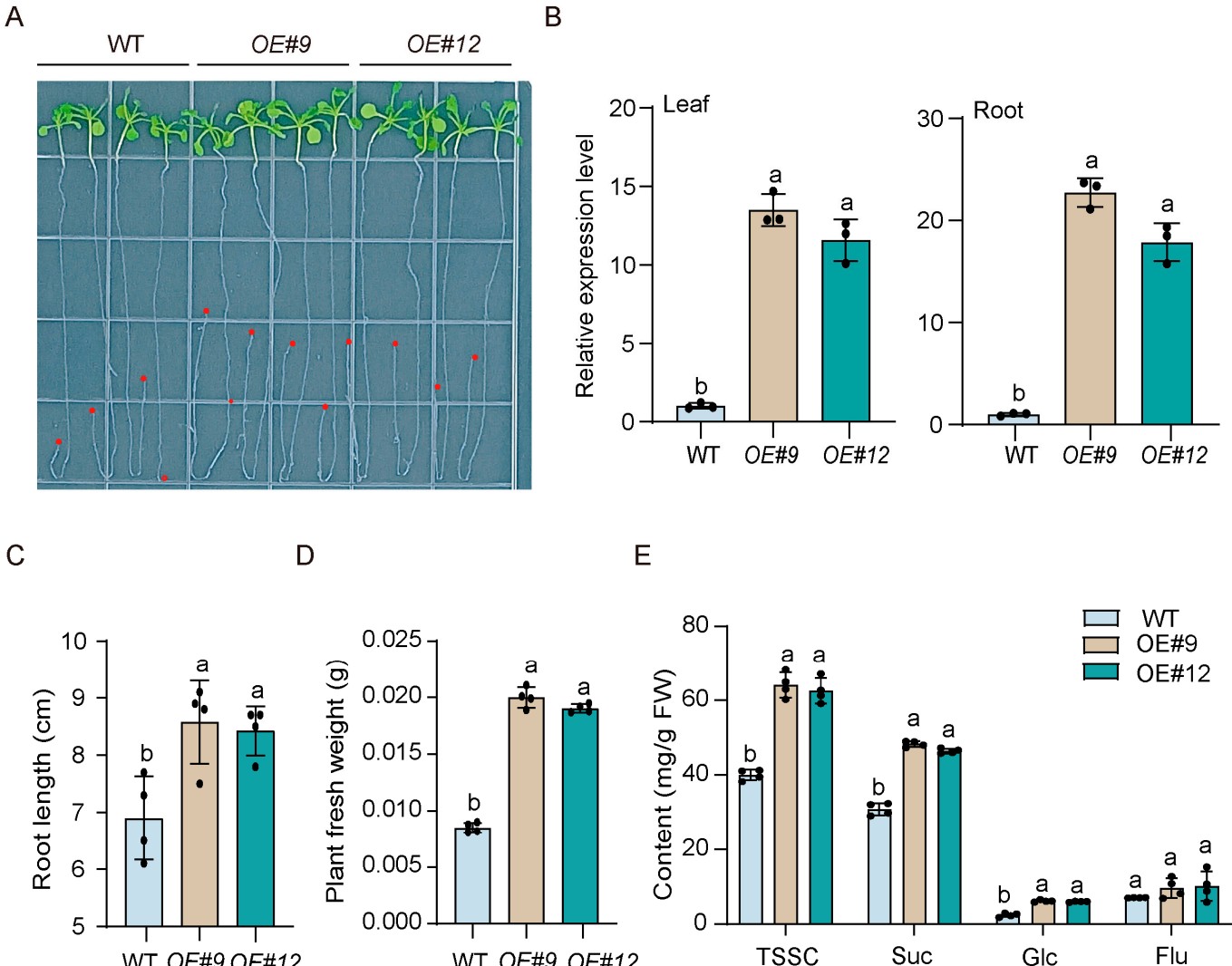

**Figure 9.** Overexpression of *RsSWEET17* gene improves the growth and development of *Arabidopsis* plants. (**A**) Phenotypes of OE-*RsSWEET17* and WT plants under MS medium for 20 days. The red dot represents the root tip of *Arabidopsis* plants. (**B**) Expression profile of the *RsSWEET17* gene in the leaves and roots of OE-*RsSWEET17* and WT plants as determined by RT-qPCR; data are mean ± SD (n = 3). (**C**) Root length of OE-*RsSWEET17* and WT plants; data are mean ± SD (n = 4). (**D**) Plant fresh weight of OE-*RsSWEET17* and WT plants; data are mean ± SD (n = 4). (**E**) The content of total soluble total sugar (TSSC), sucrose (Suc), glucose (Glc) and fructose (flu) in OE-*RsSWEET17* and WT plants; data are mean ± SD (n = 4). Different letters indicate significant differences at $p < 0.05$ according to Duncan's multiple range test.

## 4. Discussion

### 4.1. Characterization of the RsSWEET Gene Family in Radish

Although the SWEET gene family has been systematically investigated in several plant species [15,16,44–46], a systematic analysis of the SWEET gene family is still lacking in radish. In this study, 33 putative *SWEET* genes were identified in the radish genome. The phylogenetic relationship indicated that most *RsSWEETs* were more similar to *BrSWEETs* than *AtSWEETs*, which is explained by the fact that *A. thaliana*, *B. rapa* and *R. sativus* belong to the Brassicaceae family [6]. In addition, the number of *SWEET* genes in radish was significantly higher than that in *Arabidopsis* [15], rice [18], pear [45] and *B. oleracea* [47], indicating that several *RsSWEET* genes might have experienced whole-genome duplication (WGD) events in the evolutionary process in radish. *RsSWEET7* (*RsSWEET7a*, *RsSWEET7b* and *RsSWEET7c*) had three orthologous genes in *B. rapa*, while two *RsSWEET1* genes

(*RsSWEET1a* and *RsSWEET1b*) had an orthologous relationship with *AtRsSWEET1*. The results indicated that the WGD might be a vital driving force that greatly contributed to the expansion of the SWEET gene family in radish [16,19]. Moreover, the number of *SWEET* gene members ordered into Clade BI was larger than that in the other three clades, suggesting that Clade III might have been extensively expanded during genome evolution to adapt to the environment and its own metabolism. Furthermore, *RsSWEET11a/b* and *RsSWEET14a/b* in Type BI contain more exons than *RsSWEET5a/b* in Type AI, indicating that variation in the exon–intron structure might play a significant role in the evolution of RsSWEET gene families [48,49], accounting for the diversification in gene structure.

*4.2. Potential Roles of RsSWEET Genes in Various Stress Responses and Plant Growth and Development*

SWEET is an important functional protein responsible for plant growth and development as well as adaptation to environmental stresses [16,23,50,51]. Comprehensive gene expression analysis of the *RsSWEET* genes has revealed that the expression patterns were distinct in different tissues of *Arabidopsis* [15], rice [18] and *B. rapa* [16]. In this study, *RsSWEET4b* and *RsSWEET5b* were abundant in the stamen, and their functions might be similar to those of *BrSWEET4a* and *BrSWEET5c*, which play a crucial role in anther development [16]. *RsSWEET11b* and *RsSWEET12a/b* were highly expressed in leaves, and these three genes were grouped in the same subfamily with *AtSWEET11* and *AtSWEET12*. Research indicated that *AtSWEET11* and *AtSWEET12* played a vital role in sucrose efflux in the leaf [6]. It can be deduced that *RsSWEET11b* and *RsSWEET12a/b* may contribute to the efflux of sucrose from the leaf and transport to the root and other tissues to improve plant growth and development. The *atsweet11/atsweet12* double mutants showed root growth reduction [52]. *RsSWEET2b*, *RsSWEET16b* and *RsSWEET17* were strongly enriched in radish roots, indicating they might significantly function in root development and yield improvement.

In addition, the *SWEET* genes were reported to be involved in the response to abiotic stresses such as drought, salinity and cold in plants [22,53,54]. *AtSWEET1* and *AtSWEET2* were induced in *Arabidopsis*, and *OsSWEET7c* and *OsSWEET15* were regulated in rice under salt stress [28]. Overexpression of *DsSWEET17* in *Arabidopsis* could improve salt tolerance [55]. It was shown that plants control osmotic and intercellular turgor to adapt to various stress conditions by regulating the direction and transport rate of soluble sugars [23]. In this study, *RsSWEET* (*2a*, *3a*, *12a*, *17*) genes were induced to reach the maximum expression level at 12 h of cold stress, while *RsSWEET16b* and *RsSWEET12b* reached the highest expression level at 7 days. Under heat stress, the maximum expression level of *RsSWEET3a* and *RsSWEET11b* occurred at 8 h, while *RsSWEET2a* and *RsSWEET12a* were upregulated and reached the highest expression level at 12 h. The results revealed that most *RsSWEETs* could be responses to multiple stresses and play a strategically manipulating role in the course of different stresses.

*4.3. RsSWEET17 Positively Regulates Sugar Accumulation and Root Growth in Radish*

*SWEET* genes as sugar transporters are involved in the transportation of sugars such as sucrose, glucose and fructose [6,52,56]. It was revealed that soluble sugar was effectively regulated by the *SWEETs* to determine yield and improve root development in various crops, such as tomato [4], rice [5] and *Arabidopsis* [57]. In this study, consistent with the results of *A. thaliana* [52] and cucumber [2], sugar transport activity analysis revealed that *RsSWEET17* supported the uptake of hexoses. *RsSWEET17* might be involved in intracellular hexose homeostasis and contribute to sink strength. In addition, a previous study revealed that a *DsSWEET17*-overexpression transgenic line exhibited longer roots and greater fresh weight compared to the wild type in *Arabidopsis* [57]. The *SWEET17* gene is mainly expressed in vascular tissue and meristem cells of the root tip, which is involved in lateral root development in *Arabidopsis* plants [23]. The cambium mediates the process of root thickening, which strongly plays an essential role in the yield of root

crops [40,58]. In the current study, *RsSWEET17* was found to be highly abundant in roots and specifically expressed in the cambium regions of the radish taproot. In addition, *RsSWEET17*-overexpression *Arabidopsis* plants showed longer roots, higher fresh weight and increased soluble sugar content. It could be concluded that *RsSWEET17* might affect the thickening of radish taproot by improving soluble sugar-mediated cambium activity.

## 5. Conclusions

In this study, a total of 33 *RsSWEET* genes distributed on 8 chromosomes were firstly identified from the radish genome. Expression profiling analysis indicated that a majority of *RsSWEET* genes exhibited tissue-specific expression in radish. Moreover, several *RsSWEET* genes (e.g., *RsSWEET2a*, *RsSWEET3a*, *RsSWEET16b* and *RsSWEET17*) exhibited dramatically differential expression profiles under various stresses, including cold, heat, salt, Cd and Pb stress. Notably, *RsSWEET17* was highly expressed in cambium regions of radish taproot, and it was found to have the ability to transport sugar. Further transformation analysis indicated that the *RsSWEET17* positively regulated the root length and fresh weight by enhancing the soluble sugar content in *Arabidopsis* plants. These results could provide a solid foundation for dissecting the molecular regulatory networks of RsSWEET-mediated sugar accumulation and plant development and stress responses in radish.

**Supplementary Materials:** The following supporting information can be downloaded at https://www.mdpi.com/article/10.3390/horticulturae9060698/s1, Figure S1: Multiple alignments for SWEET protein sequences of *A. thaliana*, *B. rapa* and *R. sativus*. Figure S2: RT-qPCR expression profiles of *RsSWEET* genes under salt, Cd and Pb stress. Data are mean ± SD (n = 3). Asterisks above bar indicate a statistically significant difference using two-sided Student's *t*-test (ns, not significant; * $p < 0.05$; ** $p < 0.01$; *** $p < 0.001$); Table S1: The name and amino acid sequence of SWEETs in *A. thaliana*, *B. rapa* and *R. sativus*; Table S2: The orthologous gene pairs of *SWEET* genes among *R. sativus*, *A. thaliana* and *B. rapa* genomes; Table S3: Type and number of element responsive to phytohormone and stresse in the upstream regions of *RsSWEET* genes; Table S4: The primers used in this study.

**Author Contributions:** Conceptualization, L.L. and X.Z.; writing—original draft preparation, X.Z. and Y.C.; methodology, X.Z. and Y.C.; software, X.Z. and R.X.; writing—review and editing, L.X., Y.W., L.W., Y.M. and L.L.; project administration, L.L. All authors have read and agreed to the published version of the manuscript.

**Funding:** This work was supported by grants from the Jiangsu Seed Industry Revitalization Project (JBGS(2021)071), the National Natural Science Foundation of China (32172579), the Jiangsu Agricultural S&T Innovation Fund (CX (2022) 3369), the earmarked fund for Jiangsu Agricultural Industry Technology System (JATS (2023)) and the Guidance Foundation of the Hainan Institute of Nanjing Agricultural University (NAUSY-MS02).

**Data Availability Statement:** Publicly available datasets were analyzed in this study. The radish genomic data can be found here: [https://ngdc.cncb.ac.cn/bioproject/browse/PRJCA011486]. The RNA-seq data can be found here: [https://ngdc.cncb.ac.cn/search/?dbId=&q=PRJCA011507], [GenBank: PRJNA475856].

**Conflicts of Interest:** The authors declare no conflict of interest.

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
