# Peer review of "Genome-Wide Identification of the RsSWEET Gene Family and Functional Analysis of RsSWEET17 in Root Growth and Development in Radish"

_horticulturae, doi:10.3390/horticulturae9060698_

Round 1

Reviewer 1 Report (Previous Reviewer 1)

Dear authors and editor;

after a second revision of the article titled "Genome-wide identification of the RsSWEET Gene family and functional analysis of RsSWEET17 in root growth and development in radish"   under read identity "horticulturae-2452368"

I give a favorable sign of acceptance given the corrections made by the authors and they have taken into consideration all my remarks and suggestions proposed in the improvement of the content of manuscipt.

Good Lucck

acceptable and the author has corrected all language remarks and I am not a language expert

Author Response

Point-by-Point Response

Manuscript ID: horticulturae-2452368

Title: Genome-wide identification of the RsSWEET Gene family and functional analysis of RsSWEET17 in root growth and development in radish

Authors: Xiaoli Zhang, Yang Cao, Ruixian Xin, Liang Xu, Yan Wang, Lun Wang, Yinbo Ma, Liwang Liu*

Received: 30 May 2023

Dear reviewer,

Many thanks for your positive and constructive comments on this manuscript (horticulturae-2452368) entitled with ‘Genome-wide identification of the RsSWEET Gene family and functional analysis of RsSWEET17 in root growth and development in radish’

Based on the suggestions and comments, we revised the manuscript carefully and made some corresponding revisions in the revised manuscript. We addressed the revisions and corrections in the following Point-by-Point Authors’ Responses to Comments.

Response to Reviewer 1:

Comments and Suggestions for Authors

Dear authors and editor;

after a second revision of the article titled "Genome-wide identification of the RsSWEET Gene family and functional analysis of RsSWEET17 in root growth and development in radish" under read identity "horticulturae-2452368"

I give a favorable sign of acceptance given the corrections made by the authors and they have taken into consideration all my remarks and suggestions proposed in the improvement of the content of the manuscript.

Good Luck

Comments on the Quality of English Language acceptable and the author has corrected all language remarks and I am not a language expert.

Response: Many thanks for the reviewer’s positive comments.

Reviewer 2 Report (Previous Reviewer 2)

  Despite the fact that the authors of the article have made all the corrections, but considering the amount of corrections made, it is recommended to consider the issues specified by the referees again before publishing the article.

Author Response

Point-by-Point Response

Manuscript ID: horticulturae-2452368

Title: Genome-wide identification of the RsSWEET Gene family and functional analysis of RsSWEET17 in root growth and development in radish

Authors: Xiaoli Zhang, Yang Cao, Ruixian Xin, Liang Xu, Yan Wang, Lun Wang, Yinbo Ma, Liwang Liu*

Received: 30 May 2023

Dear editor,

Many thanks for the editor’s and reviewers’ positive and constructive comments on this manuscript (horticulturae-2452368) entitled with ‘Genome-wide identification of the RsSWEET Gene family and functional analysis of RsSWEET17 in root growth and development in radish’

Based on the suggestions and comments, we revised the manuscript carefully and made some corresponding revisions in the revised manuscript. We addressed the revisions and corrections in the followed Point-by-Point Authors’ Response to Comments.

Response to Reviewer 2:

Comments and Suggestions for Authors

Despite the fact that the authors of the article have made all the corrections, but considering the amount of corrections made, it is recommended to consider the issues specified by the referees again before publishing the article.

Response: Many thanks for the reviewer’s positive comments. We have addressed all the issues raised by the referees in Response to Reviewers, and made the corresponding revisions in this revised manuscript.

Reviewer 3 Report (Previous Reviewer 3)

This paper is a revised version of another paper that was previously reviewed by me and rejected by the editor. In my previous review, I indicated that it was unacceptable that the data supporting the conclusions of the article were not publicly available.

This version of the manuscript states that the genomic sequence data were obtained from reference 30. It is important to note that reference number 30 does not include an accession number in its text, which would facilitate retrieval of the data. It is also important to note that reference number 30 is incompletely cited in the reference list, as it lacks information on the volume and pages numbers. 

From previous rounds of evaluation, we know that this article's data is deposited at the National Genomic Data Centre with accession PRJCA011486. It is absolutely essential that this accession number is explicitly stated in the text of the submission. This reviewer is certain that the authors have deliberately breached their obligation to make their data publicly available, both in the original genome publication and in other follow-up publications, as their data were only released few days ago, on May 26, 2023.

The new version of the manuscript also states that the RNA-seq data are derived from two studies, one related to accession number PRJCA011507 and another described in reference number 40. It is striking that the authors gave two different accession numbers, PRJCA011486 and PRJNA822518,  in their point-by-point response to the previous version, which does not instil much confidence.

Again, please note that the accession number PRJCA011507 was only released on May 27, 2023. Reference 40 contains an accession number, PRJNA475856, but it is different from PRJNA822518. PRJNA475856 seems to be the correct one and is available in both the NCBI and NGDC database. This accession number should also be given explicitly in the text.

This is only the tip of the iceberg. Only after the data are fully accessible to the readers of the text, including the evaluators, it will be possible to evaluate the manuscript in a serious manner. Please keep in mind that, as indicated in my former reports, most of the data presented do not go beyond what was already known when the genome was sequenced.

Author Response

Point-by-Point Response

Manuscript ID: horticulturae-2452368

Title: Genome-wide identification of the RsSWEET Gene family and functional analysis of RsSWEET17 in root growth and development in radish

Authors: Xiaoli Zhang, Yang Cao, Ruixian Xin, Liang Xu, Yan Wang, Lun Wang, Yinbo Ma, Liwang Liu*

Received: 30 May 2023

Dear reviewer,

Many thanks for your comments and constructive suggestions on this manuscript (horticulturae-2452368) entitled with ‘Genome-wide identification of the RsSWEET Gene family and functional analysis of RsSWEET17 in root growth and development in radish’

Based on the suggestions and comments, we revised the manuscript carefully and made some corresponding revisions in the revised manuscript. We addressed the revisions and corrections in the followed Point-by-Point Authors’ Response to Comments.

Response to Reviewer 3:

Comments and Suggestions for Authors

Point 1: This paper is a revised version of another paper that was previously reviewed by me and rejected by the editor. In my previous review, I indicated that it was unacceptable that the data supporting the conclusions of the article were not publicly available.

This version of the manuscript states that the genomic sequence data were obtained from reference 30. It is important to note that reference number 30 does not include an accession number in its text, which would facilitate retrieval of the data. It is also important to note that reference number 30 is incompletely cited in the reference list, as it lacks information on the volume and pages numbers.

Response 1: Many thanks for the reviewer’s constructive suggestion. The Bioproject number: PRJCA011486 was provided in the ‘Data availability statement’ section of reference No. 30 (Page 1002 in PBJ, 2023, 21, 990-1004).

In addition, we updated the information on the volume and pages numbers of reference No. 30 in the revised reference list as follows.

‘Xu, L.; Wang, Y.; Dong, J.H.; Zhang, W.; Tang, M.J.; Zhang, W.; Wang, K.; Chen, Y.L.; Zhang, X.L; He, Q.; Zhang, X.Y.; Wang, K.; Wang, L.; Ma, Y.; Xia, K.; Liu, L.W. A chromosome-level genome assembly of radish (Raphanus sativus L.) reveals insights into genome adaptation and differential bolting regulation. Plant Biotechnol J. 2023.’ was revised as

‘Xu, L.; Wang, Y.; Dong, J.H.; Zhang, W.; Tang, M.J.; Zhang, W.; Wang, K.; Chen, Y.L.; Zhang, X.L; He, Q.; Zhang, X.Y.; Wang, K.; Wang, L.; Ma, Y.; Xia, K.; Liu, L.W. A chromosome-level genome assembly of radish (Raphanus sativus L.) reveals insights into genome adaptation and differential bolting regulation. Plant Biotechnol J. 2023, 21, 990-1004.’ (Page 16, Line 565)

Point 2: From previous rounds of evaluation, we know that this article's data is deposited at the National Genomic Data Centre with accession PRJCA011486. It is absolutely essential that this accession number is explicitly stated in the text of the submission. This reviewer is certain that the authors have deliberately breached their obligation to make their data publicly available, both in the original genome publication and in other follow-up publications, as their data were only released few days ago, on May 26, 2023.

Response 2: Many thanks for the reviewer’s constructive suggestions. We had added the accession number in the revised manuscript. Accordingly, ‘The sequences of radish SWEET proteins were downloaded from the ‘NAU-LB’ radish genome [30].’ was revised as:

‘The sequences of radish SWEET proteins were downloaded from the ‘NAU-LB’ radish genome (Bioproject number: PRJCA011486) [30].’

We were so sorry for the delay of genome data release. Indeed, we initially submitted the genome sequences under the Bioproject number PRJCA011486 to the National Genomics Data Center (NGDC) database (https://ngdc.cncb.ac.cn) on August 27, 2022. However, until the article was published, we found that some issues related to the gff annotation files were raised after validation check processing from the Genome Warehouse (GWH) team of NGDC. We took a very long time to check and fix each error to ensure the data accuracy. Finally, the updated genome date was successfully released into the NGDC database on May 26, 2023.

Point 3: The new version of the manuscript also states that the RNA-seq data are derived from two studies, one related to accession number PRJCA011507 and another described in reference number 40. It is striking that the authors gave two different accession numbers, PRJCA011486 and PRJNA822518, in their point-by-point response to the previous version, which does not instill much confidence.

Again, please note that the accession number PRJCA011507 was only released on May 27, 2023. Reference 40 contains an accession number, PRJNA475856, but it is different from PRJNA822518. PRJNA475856 seems to be the correct one and is available in both the NCBI and NGDC database. This accession number should also be given explicitly in the text.

Response 3: Many thanks for the reviewer’s constructive suggestions.

The RNA-seq data of four different tissues (leaf, root, pistil and stamen) was deposited in National Genomics Date Center (NGDC) under the Bioproject number PRJCA011507. In addition, the RNA-seq data of three tissues and stages (5/7/9 weeks) about radish taproot could be found under the GenBank Accession number PRJNA475856 (in Reference No. 40). The radish genome sequences were deposited in the NGDC database under the Bioproject number PRJCA011486.

We are sorry for error in citing PRJCA011486 and PRJNA822518 in our previous point-by-point response, for PRJCA011486 is Bioproject number for genome sequences in NGDC. The right accession number in Reference No. 40 is PRJNA475856, and we added this accession number instead of PRJNA822518 in the revised manuscript.

 ‘Previous RNA-seq data was retrieved to analyse the expression level of the RsSWEET genes in four different tissues (leaf, root, pistil and stamen) (BioProject number: PRJCA011507, National Genomics Date Center) [30] and in three tissues (Ca: Cambium, Co: Cortex, Pa: Parenchyma) and stages (5/7/9 weeks) of radish taproot [40].’ was revised as

‘Previous RNA-seq data was retrieved to analyse the expression level of the RsSWEET genes in four different tissues (leaf, root, pistil and stamen) (BioProject number: PRJCA011507, National Genomics Date Center) [30] and in three tissues (Ca: Cambium, Co: Cortex, Pa: Parenchyma) and stages (5/7/9 weeks) of radish taproot (GenBank: PRJNA475856) [40].’

Point 4: This is only the tip of the iceberg. Only after the data are fully accessible to the readers of the text, including the evaluators, it will be possible to evaluate the manuscript in a serious manner. Please keep in mind that, as indicated in my former reports, most of the data presented do not go beyond what was already known when the genome was sequenced.

Response 4: Yes, the related data should be publicly accessible when the article was published. Currently, in this study, the ‘NAU-LB’ radish genome date had been publicly accessible under the Bioproject number PRJCA011486 in the NGDC database (https://ngdc.cncb.ac.cn/bioproject/browse/PRJCA011486). Many thanks for the reviewer’s comments and suggestions on this Manuscript.

This manuscript is a resubmission of an earlier submission. The following is a list of the peer review reports and author responses from that submission.

Round 1

Reviewer 1 Report

Dear authors,

after review of the article entitled "Genome-wide identification of the RsSWEET Gene family and functional analysis of RsSWEET17 in root growth and development in radish" under the reference horticulturae-2376659 I give a favorable opinion given its importance and the methodology is well described .

I ask to improve the quality of the figures and to describe more the objective of the work in the introduction

all the best

Reviewer 2 Report

In the manuscript entitled “Genome-wide identification of the RsSWEET Gene family and functional analysis of RsSWEET17 in root growth and development in radish" the main objective of the authors was to identify and molecular function analysis of SWEET genes in radish. The study was good planned and performed. However, some minor issues should be clarified as outlined below before publication of the manuscript.

Line 104: Is this period of stress enough for salinity stress? And the plants experienced real salt stress?

 Line 245: Which criteria were used to select the genes?

Line 263: A and B sections as well as Cold stress are not shown on the figure and its caption.

Conclusion: In this section author are advised to show more major findings of their study.

Reviewer 3 Report

I have carefully read the manuscript and, while the manuscript is understandable, I came across abundant errors that made it difficult to follow. I have compiled a list of such errors (see below) to help the authors to fix them. The manuscript addresses the characterization of the SWEET gene/protein family in radish. Overall, the novelty of the work is low, as this family has already been characterized in other species. The sequences used in this manuscript have been taken from the genome annotation, but my understanding is that no additional efforts to validate these structure experimentally (beyond bioinformatics) have been attempted. For the sake of reproducibility the authors should deposit all sequences and raw data in public databases, or (if indicate the accession numbers if they have already been deposited). For the same reason, the authors should also be more specific and give additional details about the bioinformatic tools used. There is ample room to improve the quality of the figures and their legends: panels are in general too small to appreciate the details, and additional information must be included (units, dilution factors, names of motifs identified, etc). The manuscript would also benefit from a discussion of the orthology/paralogy relationship between the genes from radish and those of other species.

In addition to the above general considerations, these are some specific details that the authors need to address.

- line 80: please provide a link to a stable database where the genome sequence and annotation can be downloaded.

- sections 2.1, 2.2 and 2.3: please provide complete information about each bioinformatic tool used in this work, including program versions, settings used and appropriate citations.

- for the NJ algorithm, please indicate which distance metric was used and how it was corrected.

-l. 113: Please deposit the RNA-seq data at an appropriate public database (SRA) and include the accession numbers in the text.

l. 116: NODAI: please clarify what it is.

l. 121: constructed? It does not seem the correct word choice.

l. 124: belongs? It does not seem the correct word choice.

l. 134: 0.2g fresh weight (which tissue? which conditions?)

Figure 1:  The tree shown is not a NJ tree. Please provide the unrooted tree with the branch lengths drawn to scale and labeled with appropriate bootstrap values. Explain how the tree was made. Please provide the multiple sequence alignment and accession numbers of all sequences as supplemental materials.

Figure 3: This figure is difficult to see, as the three sections are too small. Please enlarge it to a reasonable size, so that  the names of the sequences, the domains and the tree can be seen without difficulty.

Section 3.3: Please clearly state how you defined the promoter regions.

Figure 7: indicate dilutions.

My understanding is that the language needs to be revised to address some problems. I hope the following list (not comprehensive) will help to authors in this task:

line 12: the first sentence of the manuscript lacks a verb. "genes essential roles" -> "genes play essential roles"

l. 15: distributed -> are distributed

l. 23: specific expressed -> specifically expressed

l. 24: heterologous expressed -> heterologously expressed

l. 25: overexpressed -> overexpressing

l. 28: taproot -> taproots

l. 29: would facilitate clarifying -> will help to clarify

l. 35: are crucial -> are a crucial

l. 42: transporter -> transporters

l. 42: be acted -> act

l. 47: MSTs transporter proteins -> MST transporter proteins

l. 47: are protein family -> are a family

l. 49: The SWEET genes -> SWEET genes

l. 54: lead into -> lead to

l. 55: evidences have -> evidence has

l. 56: genes also involved in response -> genes are also involved in the response

l. 60: that the SWEET genes play important role -> that SWEET genes play important roles

l. 64: and transported to sink organ -> and are transported to sink organs

l. 66: remains -> remain

l. 70: profile ... are -> profiles are

l. 74:  findings would provide novel insights into molecular basis -> findings provide novel insights into the molecular basis

l. 76: facilitate -> will facilitate

l. 100: Radish plant ... was -> Radish plants ... were

l. 108:  constructs ... was -> constructs were.
Please clarify how many constructs were used (or change the sentence to  singular if it was only one)

l. 133: The soluble sugars -> Soluble sugars

l. 136: sugar -> sugars

l. 139: as -> was

l. 151: obtained from -> identified in the

l. 152: homology with -> similarity to

l. 153: namely as subfamily (remove)

l. 160: belonged -> belonging

l. 176: intron-exon patterns -> intron-exon structure

l. 177: genetic structure -> gene structure

l. 219: in BI group -> in the BI group

l. 220: were expressed in stamen with a high level -> were highly expressed in stamens

l. 222: stamen but rarely -> stamens but were rarely

l. 222: root -> roots

l. 222: slightly difference -> slight difference

l. 225: this gene -> these genes (the sentence refers to two genes)

and more...

Author Response

Point-by-Point Response

Manuscript ID: horticulturae-2376659

Title: Genome-wide identification of the RsSWEET Gene family and functional analysis of RsSWEET17 in root growth and development in radish

Authors: Xiaoli Zhang, Yang Cao, Ruixian Xin, Liang Xu, Yan Wang, Lun Wang, Yinbo Ma, Liwang Liu*

Received: 17 April 2023

Dear editor,

Many thanks for the editor’s and reviewers’ positive and constructive comments on this manuscript (horticulturae-2376659) entitled with ‘Genome-wide identification of the RsSWEET Gene family and functional analysis of RsSWEET17 in root growth and development in radish’

Based on the suggestions and comments, we revised the manuscript carefully and made some corresponding revisions in the revised manuscript. We have addressed the revisions and corrections in the followed Point-by-Point Authors’ Response to Comments.

Response to Reviewer 3:

Some punctual requests are:

Point 1: Overall, the novelty of the work is low, as this family has already been characterized in other species.

Response 1: Many thanks for the reviewer’s constructive suggestion. Radish (Raphanus sativus) is an economically important root vegetable with high nutritional value. The taproot makes up an essential part of the radish diet. The accumulation of carbohydrates is beneficial for root development [31]. Carbon assimilates are products of photosynthesis and are transported to sink organ to support plant development. SWEETs (sugars will eventually be exported transporters), which play crucial roles in multiple processes, including carbohydrate transportation and plant growth and development [6]. It is feasible to improve taproot development through optimizing sucrose allocation with genetic manipulation of the SWEET gene. Therefore, the systematical identification and functional analysis of RsSWEET genes is of great importance in radish

The relevant novelty of this study could be reflected in the following aspects:

  The RsSWEET genes family was firstly characterized at the whole genome level in radish. Secondly, the RsSWEET17 had the ability to transport sugar in yeast, and it was highly expressed in radish taproot cambium. Thirdly, transgenic analysis revealed that RsSWEET17 positively regulates radish taproot development by strategically manipulating sugar accumulation.

Some corresponding revisions were performed in the conclusion section as follows.

‘In this study, a total of 33 RsSWEET genes were firstly identified from radish genome, which were distributed to eight radish chromosomes. Expression profiling analysis indicated that a majority of RsSWEET genes exhibited tissue-specific expression in radish. Moreover, several RsSWEET genes (e.g. RsSWEET2a, RsSWEET3a, RsSWEET16b and RsSWEET17) exhibited dramatically differential expression profiles under various stresses including cold, heat, salt, Cd and Pb stress. Notably, the RsSWEET17 was highly expressed in cambium regions of radish taproot, and it was identified to have the ability to transport sugar. Further transformation analysis indicated that the RsSWEET17 positively regulated the root length and fresh weight by enhancing the soluble sugar content in Arabidopsis plants. These results could provide the solid foundation for dissecting the molecular regulatory networks of RsSWEET-mediated sugar accumulation and plant development and stress response in radish.’ (Page 12, lines 48-51; Page 13, lines 1-9)

Point 2: The sequences used in this manuscript have been taken from the genome annotation, but my understanding is that no additional efforts to validate these structure experimentally (beyond bioinformatics) have been attempted.

Response 2: Thanks for the reviewer’s suggestion. Firstly, the protein domain of SWEET (PF03383) obtained from the Pfam database was applied to search the NAU-LB radish genome database. Then, the conserved domains of the SWEET were further verified using the SMART (https://smart.embl.de/) and Pfam (http://pfam-legacy.xfam.org/). Finally, a total of 33 SWEET proteins were obtained from radish genome. Most sequences of RsSWEET genes were isolated and sequenced with Sanger method. Moreover, phylogenetic relationship of A. thaliana, B.rapa and R. sativus indicated that most RsSWEETs were more similar to BrSWEETs, which corresponds with the fact that A. thaliana, B.rapa and R. sativus belong to the Burassicaceae family. These results indicated that the gene structures of these RsSWEETs were highly conserved. Similar results were also reported in some other previous study (Miao et al., 2018). Due to the experimental validation of gene structure takes a relatively long period, we would carry out this work in the near future.

References

Miao, L.; Lv, Y.; Kong, L.; Chen, Q.; Chen, C.; Li, J.; Zeng, F.; Wang, S.; Li, J.; Huang, L.; Cao, J.; Yu, X. Genome-wide identification, phylogeny, evolution, and expression patterns of MtN3/saliva/SWEET genes and functional analysis of BcNS in Brassica rapa. BMC Genomics. 2018, 19, 174.

Point 3: For the sake of reproducibility the authors should deposit all sequences and raw data in public databases, or (if indicate the accession numbers if they have already been deposited).

Response 3: Many thanks for the reviewer’s constructive suggestion. In this study, the sequences of radish SWEET proteins were downloaded from the ‘NAU-LB’ radish genome (Xu et al., 2023, BioProject number: PRJCA011486).

The RNA-seq data used in this study was retrieved to analyse the expression level of the RsSWEET genes in four different tissues (leaf, root, pistil and stamen) (Xu et al., 2023) and in three tissues (Ca: Cambium, Co: Cortex, Pa: Parenchyma) and stages (5/7/9 weeks) of radish taproot (Hoang et al., 2020). Some corresponding revisions were performed in this revised manuscript. (Page 3, lines 30-34)

The SWEET protein sequences and gene accession numbers of A. thaliana, B.rapa and R. sativus were provided in supplementary table S1. These SWEET genes accession numbers of A. thaliana, B.rapa and R. sativus were performed as follows.

Arabidopsis thaliana

Brassica rapa

Raphanus sativus

Gene ID

Gene name

Gene ID

Gene name

Gene ID

Gene name

AT1G21460

AtSWEET1

BraA08004869

BrSWEET1a

Rsa8g008140

RsSWEET1a

AT3G14770

AtSWEET2

BraA06001748

BrSWEET1b

Rsa1g014650

RsSWEET1b

AT5G53190

AtSWEET3

BraA01002509

BrSWEET2

Rsa5g009870

RsSWEET2a

AT3G28007

AtSWEET4

BraA02001501

BrSWEET3a

Rsa6g006940

RsSWEET2b

AT5G62850

AtSWEET5

BraA10000924

BrSWEET3b

Rsa2g022970

RsSWEET3a

AT1G66770

AtSWEET6

BraA06005558

BrSWEET4a

Rsa9g013830

RsSWEET3b

AT4G10850

AtSWEET7

BraA02002561

BrSWEET4b

Rsa9g014560

RsSWEET3c

AT5G40260

AtSWEET8

BraA09002123

BrSWEET5a

Rsa4g020440

RsSWEET4a

AT2G39060

AtSWEET9

BraA02003246

BrSWEET5b

Rsa6g021400

RsSWEET4b

AT5G50790

AtSWEET10

BraA06000185

BrSWEET7

Rsa9g044840

RsSWEET5a

AT3G48740

AtSWEET11

BraA04001505

BrSWEET8a

Rsa6g016720

RsSWEET5b

AT5G23660

AtSWEET12

BraA05003946

BrSWEET8b

Rsa5g045530

RsSWEET5c

AT5G50800

AtSWEET13

BraA03002083

BrSWEET9

Rsa6g016700

RsSWEET5d

AT4G25010

AtSWEET14

BraA03001595

BrSWEET10a

Rsa9g024060

RsSWEET7a

AT5G13170

AtSWEET15

BraA07000853

BrSWEET10b

Rsa9g023450

RsSWEET7b

AT3G16690

AtSWEET16

BraA06004314

BrSWEET11a

Rsa3g003240

RsSWEET7c

AT4G15920

AtSWEET17

BraA06001603

BrSWEET11b

Rsa4g033550

RsSWEET8

BraA03004253

BrSWEET11c

Rsa3g011990

RsSWEET9

BraA06004941

BrSWEET12a

Rsa3g008890

RsSWEET10

BraA09002020

BrSWEET12b

Rsa4g015190

RsSWEET11a

BraSca000162

BrSWEET13

Rsa4g015200

RsSWEET11b

BraA08004017

BrSWEET14a

Rsa5g040710

RsSWEET12a

BraA03006198

BrSWEET14b

Rsa6g041060

RsSWEET12b

BraA02000477

BrSWEET15a

Rsa9g016910

RsSWEET13a

BraA08001708

BrSWEET15b

Rsa2g009170

RsSWEET13b

BraA03000581

BrSWEET15c

Rsa2g039010

RsSWEET14a

BraA01002377

BrSWEET16a

Rsa4g007650

RsSWEET14b

BraA03003806

BrSWEET16b

Rsa8g015330

RsSWEET14c

BraA03005641

BrSWEET17

Rsa2g028360

RsSWEET15a

Rsa3g026620

RsSWEET15b

Rsa5g050340

RsSWEET16a

Rsa6g008090

RsSWEET16b

Rsa4g012210

RsSWEET17

References

Xu, L.; Wang, Y.; Dong, J.H.; Zhang, W.; Tang, M.J.; Zhang, W.; Wang, K.; Chen, Y.L.; Zhang, X.L; He, Q.; Zhang, X.Y.; Wang, K.; Wang, L.; Ma, Y.; Xia, K.; Liu, L.W. A chromosome-level genome assembly of radish (Raphanus sativus L.) reveals insights into genome adaptation and differential bolting regulation. Plant Biotechnol J. 2023.

Hoang, N.V.; Choe, G.; Zheng, Y.; Aliaga Fandino, A.C.; Sung, I.; Hur, J.; Kamran, M.; Park, C.; Kim, H.; Ahn, H.; Kim, S.; Fei, Z.; Lee, J.Y. Identification of Conserved Gene-Regulatory Networks that Integrate Environmental Sensing and Growth in the Root Cambium. Curr Biol. 2020, 30, 2887-2900.

Point 4: For the same reason, the authors should also be more specific and give additional details about the bioinformatic tools used.

Response 4: Many thanks for the reviewer’s constructive suggestion. In this revised manuscript, we revised the corresponding sentence in 2.1-2.3 section as follows.

‘To confirm the predicted genes, the seed sequence (PF03383, http://pfam.xfam.org/) with MtN3/saliva domain was searched against the radish genome using the hidden Markov model (HMM) search tool with an E-value set to 0.01 [29,30].’ was revised as

‘To confirm the predicted genes, the seed sequence (PF03383, http://pfam.xfam.org/) with MtN3/saliva domain was searched against the radish genome using HMMER 3.0 and BLASTP program with an E-value set to 0.01 [31,32].’ (Page 2, lines 40-45)

‘MUSCLE was used to perform multiple alignments of the SWEET protein sequences of Arabidopsis, B.rapa and radish with default parameters. Then, dendrograms were constructed by MEGA 6.0 via the neighbor-joining (NJ) method and bootstrap 1000 replicates [31]. In addition, RsSWEET genes were assigned to eight chromosomes by TBtools software based on the corresponding location parameters in the NAU-LB radish genome [28]. The Gene Structure Display Server (GSDS) was used to analyze intron-exon structure of each gene [32]. The MEME tool was employed to identify conserved motifs [33].’ was revised as

‘The SWEET protein sequence of Arabidopsis was downloaded from TAIR10 (https://www.arabidopsis.org/). The SWEET protein sequence of Brassica rapa was obtained from Brassica database (https://www.genoscope.cns.fr/brassicanapus/). MUSCLE was used to perform multiple alignments of the full-length SWEET protein sequences of A. thaliana, B.rapa and R. sativus with default parameters. Subsequently, the phylogenetic tree was constructed by MEGA 6.0 via the neighbor-joining (NJ) method with bootstrap 1000 replicates and the Poisson model [34,35]. In addition, RsSWEET genes were mapped to eight chromosomes by TBtools software based on the corresponding location parameters in the NAU-LB radish genome [30]. The Gene Structure Display Server (GSDS) (http://gsds.gao-lab.org/) was used to analyze intron-exon structure of each gene [36]. The MEME tool (https://meme-suite.org/meme/doc/meme.html) was employed to identify conserved motifs [37].’ (Page 3, lines 1-10)

Point 5: There is ample room to improve the quality of the figures and their legends: panels are in general too small to appreciate the details, and additional information must be included (units, dilution factors, names of motifs identified, etc).

Response 5: Many thanks for the reviewer’s constructive comments. We have improved these figures in this revised manuscript.

Figure 1(Page 5, lines 1),

Figure 2(Page 5, lines 4),

Figure 3(Page 6, lines 9),

Figure 4(Page 7, lines 1),

Figure 6(Page 9, lines 3)

Figure 7(Page 10, lines 1) and Figure 8(Page 11, lines 1) were updated in revised manuscript.

Point 6: The manuscript would also benefit from a discussion of the orthology/paralogy relationship between the genes from radish and those of other species.

Response 6: Many thanks for the reviewer’s constructive comments. In this study, we mainly focused on the identification of all RsSWEET genes from the radish genome, and explored the phylogenetic relationships, gene structures, conserved domains and cis-acting elements of RsSWEET gene family in radish. In addition, the expression profiles of RsSWEET genes are investigated in different tissues and developmental stages, and the differential expression of eight RsSWEET genes (RsSWEET2a, RsSWEET3a, RsSWEET11a/b, RsSWEET12a/b, RsSWEET16b, and RsSWEET17) were assessed under various abiotic stresses with RT-qPCR analysis. Furthermore, the potential roles of RsSWEET17 in sugar accumulation and taproot development were investigated by heterologous expressed in yeast strain EBY.VW4000 and genetic manipulation in radish.

Considering the complexity of the orthology/ paralogy analysis and the limited time for revising manuscript, we would carry out investigation of the orthology/ paralogy relationship between the genes from radish and those of other species in next paper in the near future.

Point 7: - sections 2.1, 2.2 and 2.3: please provide complete information about each bioinformatic tool used in this work, including program versions, settings used and appropriate citations.

Response 7: Many thanks for the reviewer’s constructive suggestion. In this revised manuscript, we revised the corresponding sentence in 2.1-2.3 section as follow.

‘To confirm the predicted genes, the seed sequence (PF03383, http://pfam.xfam.org/) with MtN3/saliva domain was searched against the radish genome using the hidden Markov model (HMM) search tool with an E-value set to 0.01 [29,30].’ was revised as

‘To confirm the predicted genes, the seed sequence (PF03383, http://pfam.xfam.org/) with MtN3/saliva domain was searched against the radish genome using HMMER 3.0 and BLASTP program with an E-value set to 0.01 [31,32].’ (Page 2, lines 40-45)

‘MUSCLE was used to perform multiple alignments of the SWEET protein sequences of Arabidopsis, B.rapa and radish with default parameters. Then, dendrograms were constructed by MEGA 6.0 via the neighbor-joining (NJ) method and bootstrap 1000 replicates [31]. In addition, RsSWEET genes were assigned to eight chromosomes by TBtools software based on the corresponding location parameters in the NAU-LB radish genome [28]. The Gene Structure Display Server (GSDS) was used to analyze intron-exon structure of each gene [32]. The MEME tool was employed to identify conserved motifs [33].’ was revised as

‘The SWEET protein sequence of Arabidopsis was downloaded from TAIR10 (https://www.arabidopsis.org/). The SWEET protein sequence of Brassica rapa was obtained from Brassica database (https://www.genoscope.cns.fr/brassicanapus/). MUSCLE was used to perform multiple alignments of the full-length SWEET protein sequences of A. thaliana, B.rapa and R. sativus with default parameters. Subsequently, the phylogenetic tree was constructed by MEGA 6.0 via the neighbor-joining (NJ) method with bootstrap 1000 replicates and the Poisson model [34,35]. In addition, RsSWEET genes were mapped to eight chromosomes by TBtools software based on the corresponding location parameters in the NAU-LB radish genome [30]. The Gene Structure Display Server (GSDS) (http://gsds.gao-lab.org/) was used to analyze intron-exon structure of each gene [36]. The MEME tool (https://meme-suite.org/meme/doc/meme.html) was employed to identify conserved motifs [37].(Page 3, lines 1-10)

‘PlantCARE software was used to analyze the cis-elements in the promoter sequences of RsSWEET genes (1.5 kb upstream of the translation start site) [34,35],’ was revised as

‘The PlantCARE (https://bioinformatics.psb.ugent.be/webtools/plantcare/html/) database was used to analyze the cis-elements in the promoter sequences of RsSWEET genes (1.5 kb upstream of the translation start site) [37].(Page 3, lines 14-15)

Point 8: - for the NJ algorithm, please indicate which distance metric was used and how it was corrected.

Response 8: Many thanks for the reviewer’s positive comments. The evolutionary history was inferred using the Neighbor-Joining method (Nguyen et al., 2014). The optimal tree with the sum of branch length = 7.86404076 is shown. The percentage of replicate trees in which the associated taxa clustered together in the bootstrap test (1000 replicates) is shown next to the branches. The tree is drawn to scale, with branch lengths (next to the branches) in the same units as those of the evolutionary distances used to infer the phylogenetic tree. The evolutionary distances were computed using the Poisson model (Tamura et al., 2013) and are in the units of the number of amino acid substitutions per site. All positions containing gaps and missing data were deleted.

Some corresponding revisions were performed in the conclusion section as follows.

‘The SWEET protein sequence of Brassica rapa was obtained from BRAD database (http://brassicadb.org/brad/). MUSCLE was used to perform multiple alignments of the full-length SWEET protein sequences of A. thaliana, B.rapa and R. sativus with default parameters. Subsequently, the phylogenetic tree was constructed by MEGA 6.0 via the neighbor-joining (NJ) method with bootstrap 1000 replicates and the Poisson model [34,35].’ (Page 3, lines 1-10)

References

Tamura, K., Stecher, G., Peterson, D., Filipski, A., Kumar, S. MEGA6: molecular evolutionary genetics analysis version 6.0. Mol. Biol. Evol. 2013, 30, 2725–2729.

Nguyen, LT., Schmidt, HA., von Haeseler, A., Minh, BQ. IQ-TREE: a fast and effective stochastic algorithm for estimating maximum-likelihood phylogenies. Mol Biol Evol. 2014, 32, 268–74.

Point 9: -l. 113: Please deposit the RNA-seq data at an appropriate public database (SRA) and include the accession numbers in the text.

Response 9: Many thanks for the reviewer’s constructive suggestion. These RNA-seq data had been deposited to the public database in the previous studies. Some corresponding revisions were performed as follows.

  ‘The RNA-seq data used in this study was retrieved to analyze the expression level of the RsSWEET genes in four different tissues (leaf, root, pistil and stamen) (BioProject number: PRJCA011486) (Xu et al., 2023) and in three tissues (Ca: Cambium, Co: Cortex, Pa: Parenchyma) and stages (5/7/9 weeks) of radish taproot (BioProject number: PRJNA822518) (Hoang et al., 2020).’ (Page 3, lines 30-34)

References

Xu, L.; Wang, Y.; Dong, J.H.; Zhang, W.; Tang, M.J.; Zhang, W.; Wang, K.; Chen, Y.L.; Zhang, X.L; He, Q.; Zhang, X.Y.; Wang, K.; Wang, L.; Ma, Y.; Xia, K.; Liu, L.W. A chromosome-level genome assembly of radish (Raphanus sativus L.) reveals insights into genome adaptation and differential bolting regulation. Plant Biotechnol J. 2023.

Hoang, N.V.; Choe, G.; Zheng, Y.; Aliaga Fandino, A.C.; Sung, I.; Hur, J.; Kamran, M.; Park, C.; Kim, H.; Ahn, H.; Kim, S.; Fei, Z.; Lee, J.Y. Identification of Conserved Gene-Regulatory Networks that Integrate Environmental Sensing and Growth in the Root Cambium. Curr Biol. 2020, 30, 2887-2900.

Point 10: l. 116: NODAI: please clarify what it is.

Response 10: Thanks for the reviewer’s constructive comments. NODAI represents a previous radish genome database (Jeong et al., 2016). However, the latest ‘NAU-LB’ radish genome sequences were used in this study (Xu et al., 2023). We therefore remove the sentence containing NODAI in this revised manuscript.

References

Y M, Kim N Jeong, Ahn B O, et al. Elucidating the triplicated ancestral genome structure of radish based on chromosome-level comparison with the Brassica genomes[J]. Theor Appl Genet, 2016, 129(7): 1357-1372.

Xu, L.; Wang, Y.; Dong, J.H.; Zhang, W.; Tang, M.J.; Zhang, W.; Wang, K.; Chen, Y.L.; Zhang, X.L; He, Q.; Zhang, X.Y.; Wang, K.; Wang, L.; Ma, Y.; Xia, K.; Liu, L.W. A chromosome-level genome assembly of radish (Raphanus sativus L.) reveals insights into genome adaptation and differential bolting regulation. Plant Biotechnol J. 2023.

Point 11: l. 121: constructed? It does not seem the correct word choice.

Response 11: Many thanks for the reviewer’s constructive suggestion. We used ‘fused’ replaced ‘constructed’.

Point 12: l. 124: belongs? It does not seem the correct word choice.

Response 12: Many thanks for the reviewer’s positive comments. We have improved this word in the revised manuscript.

‘belongs to’ was revise as ‘is’

Point 13: l. 134: 0.2g fresh weight (which tissue? which conditions?)

Response 13: Many thanks for the reviewer’s constructive suggestion. 0.2g fresh weight, which means ‘the fresh weight of the whole plant of Arabidopsis thaliana

The WT and transgenic Arabidopsis plants were collected to determine soluble sugar content after grown for 20d. In this revised manuscript, we revised the corresponding sentence as follows.

In brief, after growing WT and transgenic Arabidopsis plants on MS solid medium for 20d, 0.2g fresh weight of Arabidopsis plants was ground into power after being frozen in liquid nitrogen. (Page 4, lines 5-7)

Point 14: Figure 1:  The tree shown is not a NJ tree. Please provide the unrooted tree with the branch lengths drawn to scale and labeled with appropriate bootstrap values. Explain how the tree was made. Please provide the multiple sequence alignment and accession numbers of all sequences as supplemental materials.

Response 14: Many thanks for the reviewer’s constructive suggestion. We have improved this word in the revised manuscript.

The unrooted tree of SWEET proteins from R. sativus, A. thaliana and B. rapa was performed Figure 1. (Page 5, lines 1)

The multiple sequence alignment of all SWEET sequences was put in supplemental materials part as figure S1. (Page 4, lines 35)

The accession numbers of all SWEET sequences as supplemental materials table S1.

‘MUSCLE was used to perform multiple alignments of the full-length SWEET protein sequences of R. sativus, A. thaliana and B. rapa with default parameters. Subsequently, the phylogenetic tree was constructed by MEGA 6.0 via the neighbor-joining (NJ) method with bootstrap 1000 replicates and the Poisson model’ (Page 3, lines 1-10)

Figure 1. Phylogenetic relationships of SWEET proteins from A. thaliana, B.rapa and R. sativus. Subfamilies are marked in different colors. At: A. thaliana, Br: B. rapa, Rs: R. sativus.

Figure S1, Multiple alignments for SWEET protein sequences of A. thaliana, B.rapa and R. sativus. The accession numbers of SWEET sequences from three plant species were provided in the Table S1.

References

Nguyen LT, Schmidt HA, von Haeseler A, Minh BQ. IQ-TREE: a fast and effective stochastic algorithm for estimating maximum-likelihood phylogenies. Mol Biol Evol. 2014, 32, 268–74.

Point 15: Figure 3: This figure is difficult to see, as the three sections are too small. Please enlarge it to a reasonable size, so that the names of the sequences, the domains and the tree can be seen without difficulty.

Response 15: Many thanks for the reviewer’s constructive suggestion. We have improved the Figure 3 in the revised manuscript. (Page 6, line 9)

Figure 3. The analysis of RsSWEET protein and gene structure. (A) Phylogenetic relationship of RsSWEET proteins; (B) Genes structure of RsSWEET genes. (C) Conserved motifs of RsSWEET proteins.

Point 16: Section 3.3: Please clearly state how you defined the promoter regions.

Response 16: Many thanks for the reviewer’s comments. The 1.5 kb sequence upstream from the start codon of RsSWEET genes was extracted as the promoter region.

Point 17: Figure 7: indicate dilutions.

Response 17: Many thanks for the reviewer’s constructive suggestion. We have improved the figure in this revised manuscript. (Page 10, line 1)

Figure 7. Sugar transport activity analysis of RsSWEET17 in EBY.VW4000 yeast cells. Transformants were grown on SD/- Ura solid medium containing 2% maltose, glucose, fructose, galactose and mannose independently for 3d.

Point 18: Comments on the Quality of English Language

My understanding is that the language needs to be revised to address some problems. I hope the following list (not comprehensive) will help to authors in this task: line 12: the first sentence of the manuscript lacks a verb. "genes essential roles" -> "genes play essential roles"; l. 15: distributed -> are distributed; l. 23: specific expressed -> specifically expressed; l. 24: heterologous expressed -> heterologously expressed;

  1. 25: overexpressed -> overexpressingl. 28: taproot -> taproots; l. 29: would facilitate clarifying -> will help to clarify; l. 35: are crucial -> are a crucial; l. 42: transporter -> transporters; l. 42: be acted -> act and more...

Response 18: Many thanks for the reviewer’s positive comments. We have extensively revised these sentences according to reviewer’s suggestions.

‘genes essential roles’ was revised as ‘genes play essential roles’ (Page 1, line 12)

‘distributed’ was revised as ‘are distributed’ (Page 1, line 15)

‘specific expressed’ was revised as ‘specifically expressed’ (Page 1, line 23)

‘heterologous expressed’ was revised as ‘heterologously expressed’ (Page 1, line 24)

‘overexpressed’ was revised as ‘overexpressing’ (Page 1, line 25)

‘taproot’ was revised as ‘taproots’ (Page 1, line 28)

‘would facilitate clarifying’ was revised as ‘will help to clarify’ (Page 1, line 29)

‘are crucial’ was revised as ‘are a crucial’ (Page 1, line 35)

‘transporter’ was revised as ‘transporters’ (Page 1, line 42)

‘be acted’ was revised as ‘act’ (Page 1, line 43)

‘MSTs transporter proteins’ was revised as ‘MST transporter proteins’ (Page 2, line 3)

‘are protein family’ was revised as ‘are a family’ (Page 2, line 4)

‘The SWEET genes ’ was revised as ‘SWEET genes’ (Page 2, line 5)

‘lead into’ was revised as ‘lead to’ (Page 2, line 14)

‘evidences have’ was revised as ‘evidence has’ (Page 2, line 17)

 ‘remains’ was revised as ‘remain’ (Page 2, line 28)

‘findings would provide novel insights into molecular basis’ was revised as ‘findings provide novel insights into the molecular basis’ (Page 2, line 38)

‘facilitate’ was revised as ‘will facilitate’ (Page 2, line 39)

‘Radish plant ... was’ was revised as ‘Radish plants ... were’ (Page 3, line 24)

 ‘The soluble sugars’ was revised as ‘Soluble sugars’ (Page 4, line 5)

‘sugar’ was revised as ‘sugars’ (Page 4, line 9)

‘as’ was revised as ‘was’ (Page 4, line 10)

‘obtained from’ was revised as ‘identified in the’ (Page 4, line 26)

‘homology with’ was revised as ‘similarity to’ (Page 4, line 27)

We deleted ‘namely as subfamily’

‘belonged’ was revised as ‘belonging’ (Page 4, line 35)

‘intron-exon patterns’ was revised as ‘intron-exon structure’ (Page 5, line 8)

‘genetic structure’ was revised as ‘gene structure’ (Page 5, line 8)

‘in BI group’ was revised as ‘in the BI group’ (Page 7, line 9)

‘were expressed in stamen with a high level’ was revised as ‘were highly expressed in stamens’ (Page 7, line 10)

‘root’ was revised as ‘roots’ (Page 7, line 12)

‘slightly difference’ was revised as ‘slight difference’ (Page 7, line 12)

‘this gene’ was revised as ‘these genes’ (Page 7, line 15)

Additionally, after all authors checked and revised the grammar throughout the whole manuscript, the manuscript was critically revised and edited by Proof-Reading-Services.com Ltd. And the Editorial certification was provided. Hopefully the language is now more acceptable for publication.

Round 2

Reviewer 3 Report

The fact that radish is a major cultivated plant does not immediately make the work "novel". One of my objections is that the 33 genes identified in the radish genome had already been identified previously when the genome annotation was performed. In their response, the authors clearly state that the sequences have been taken from the already available sequence of the radish genome. In this sense the work is not entirely "novel" because these sequences had already been described previously, indeed by the same authors.

My view is that the publication of a new article characterizing the genes of this family requires some extra effort, beyond the one that had already been done when the genome was published, i.e. to validate the intron-exon structure of the genes. This might be carried out using the available RNA-seq data, or the sequences that the authors claim to have obtained by the Sanger method. Claiming that this would require a long time does not seem a valid reason, particularly if the authors intend to present their characterization of the gene family as novel work.

One of the authors' goals was to explore the phylogenetic relationships of the family members. For this reason, this reviewer does not understand why the paralogy and orthology relationships of the different family members have not been discussed, especially when the phylogenetic relationships between the Arabidopsis, radish and Brassica genes are presented in Figure 1. The authors could rely on the phylogenetic tree, pairwise sequence comparisons and synteny/colinearity to discuss these relationships. Again, the fact that establishing such relationships requires some extra work does not seem a valid reason for not doing so.

I am somewhat surprised that the authors give hyperlinks from which to download the Brassica and Arabidopsis sequences, but fail to do the same for the radish sequences. For the latter, only an accession number corresponding to an unnamed database is given. This accession number seems to correspond to the National Genomics Data Center, a database in China. However, although the same accession number has been published by the authors in at least three other papers (https://www.sciencedirect.com/science/article/abs/pii/S0981942822004715 , https://academic.oup.com/jxb/article-abstract/74/1/233/6761080 , and https://onlinelibrary.wiley.com/doi/full/10.1111/pbi.14011 ), this entry is not available in the corresponding database and, consequently, the sequences are not publicly available. This way of proceeding reflects a pattern of action that, in the opinion of this reviewer, represents a bad scientific practice.

The editor can check the (un)availability of the data at this link: https://ngdc.cncb.ac.cn/search/?dbId=&q=PRJCA011486

My recommendation is that the article is not be published at least until the authors demonstrate that they wish to share the sequence data for this and their previous articles, making the genome sequence and its annotation available to the public through an acceptable repository, such as SRA (raw data) and Genbank (assembled sequences), as stated in https://www.mdpi.com/journal/horticulturae/instructions#suppmaterials , in line with Horticulturae's clear instructions for authors.